# Development of CuO Nanoparticles from the Mucus of Garden Snail *Cornu aspersum* as New Antimicrobial Agents

**DOI:** 10.3390/ph17040506

**Published:** 2024-04-15

**Authors:** Pavlina Dolashka, Karina Marinova, Petar Petrov, Ventsislava Petrova, Bogdan Ranguelov, Stella Atanasova-Vladimirova, Dimitar Kaynarov, Ivanka Stoycheva, Emiliya Pisareva, Anna Tomova, Angelina Kosateva, Lyudmila Velkova, Aleksandar Dolashki

**Affiliations:** 1Institute of Organic Chemistry with Center for Phytochemistry, Bulgarian Academy of Sciences, 1000 Sofia, Bulgaria; pda54@abv.bg (P.D.); karina.marinova@orgchm.bas.bg (K.M.); petar.petrov@orgchm.bas.bg (P.P.); mitkokaynarov@abv.bg (D.K.); stoycheva@orgchm.bas.bg (I.S.); angelina.kosateva@orgchm.bas.bg (A.K.); lyudmila.velkova@orgchm.bas.bg (L.V.); 2Faculty of Biology (SU-BF), Sofia University “St. Kliment Ohridski”, 1504 Sofia, Bulgaria; vpetrova@biofac.uni-sofia.bg (V.P.); episareva@uni-sofia.bg (E.P.); aatomova@biofac.uni-sofia.bg (A.T.); 3Institute of Physical Chemistry “Rostislav Kaishev”, Bulgarian Academy of Sciences, 1000 Sofia, Bulgaria; rangelov@ipc.bas.bg (B.R.); statanasova@ipc.bas.bg (S.A.-V.)

**Keywords:** snail *Cornu aspersum*, copper nanoparticles, antimicrobial activity

## Abstract

Several biologically active compounds involved in the green synthesis of silver and gold nanoparticles have been isolated from snail mucus and characterized. This paper presents a successful method for the application of snail mucus from *Cornu aspersum* as a bioreducing agent of copper sulfate and as a biostabilizer of the copper oxide nanoparticles (CuONPs-Muc) obtained. The synthesis at room temperature and neutral pH yielded nanoparticles with a spherical shape and an average diameter of 150 nm. The structure and properties of CuONPs-Muc were characterized using various methods and techniques, such as ultraviolet–visible spectroscopy (UV–vis), high-performance liquid chromatography (HPLC), one-dimensional polyacrylamide gel electrophoresis (1D-PAGE), up-conversion infrared spectroscopy Fourier transform (FTIR), scanning electron microscopy combined with energy dispersive spectroscopy (SEM/EDS), Raman spectroscopy and imaging, thermogravimetric analysis (TG-DSC), etc. Mucus proteins with molecular weights of 30.691 kDa and 26.549 kDa were identified, which are involved in the biogenic production of CuONPs-Muc. The macromolecular shell of proteins formed around the copper ions contributes to a higher efficiency of the synthesized CuONPs-Muc in inhibiting the bacterial growth of several Gram-positive (Bacillus subtilis NBIMCC2353, *Bacillus spizizenii* ATCC 6633, *Staphylococcus aureus* ATCC 6538, *Listeria innocua* NBIMCC8755) and Gram-negative (*Escherichia coli* ATCC8739, *Salmonella enteitidis* NBIMCC8691, *Salmonella typhimurium* ATCC 14028, *Stenotrophomonas maltophilia* ATCC 17666) bacteria compared to baseline mucus. The bioorganic synthesis of snail mucus presented here provides CuONPs-Muc with a highly pronounced antimicrobial effect. These results will expand knowledge in the field of natural nanomaterials and their role in emerging dosage forms.

## 1. Introduction

The synthesis of metal nanoparticles is attracting increasing interest [1,2]. Applied physicochemical methods have several disadvantages related to the release of toxic and harmful chemicals [3] as well as the use of complex equipment and a large amount of energy [4]. The solution to this problem is the application of a new method—green synthesis of metal nanoparticles, which is an ecological method that requires less energy and ecological solvents (water, ethanol, etc.) and does not generate toxic waste. Therefore, it has emerged as an attractive research area in recent years [5,6].

Various types of metal oxide nanoparticles (silver, gold, zinc, strontium, copper, etc.) with unique physical and chemical characteristics are used as adsorbents [7].

Silver (AgNPs) and gold (AuNPs) nanoparticles synthesized using plant extracts are of great interest to researchers [8,9]. Recently, Castillo-Henríquez et al. (2020) reported the green synthesis of gold and silver nanoparticles from plant extracts and their capacity as antimicrobial agents in the field of agriculture to combat bacterial and fungal pathogens. Moreover, this work makes a brief review of nanoparticles’ contribution to water treatment and the development of “environmentally-friendly” nanofertilizers, nanopesticides, and nanoherbicides [8]. Lubis et al. [10] presented the use of *Persicaria odorata* leaf extract to reduce silver ions to AgNPs, while AuNPs were synthesized using an aqueous extract of *Limoniaacidissima* leaves, which have been shown to have an enhanced antimicrobial effects and to help in wound healing [11].

Strontium-based nanoparticles [12], as well as zinc oxide nanoparticles (ZnNPs), which can be used for wound healing as an adjuvant drug without toxic side effects, also have a strong regenerative effect on bone damaged by chemotherapeutic drugs [13]. Copper nanoparticles (CuNPs) also support the wound healing process and find biomedical applications because of their anticancer, antioxidant, anti-inflammatory, and neuroprotective effects [14,15]. Green biosynthesis of *Phragmanthera austroarabica* extract-derived CuONPs with highly effective antibacterial and antifungal effects against *Staphylococcus aureus*, *Escherichia coli*, *Penicillium chrysogenum,* and *Fusarium oxysporum* has been reported [16]. Biogenic methodologies used for the synthesis of nanoparticles are mostly limited to microorganisms and plant extracts, with very few studies conducted on body secretions of living animals. Data have been published on much more suitable sources than plants [11] for the synthesis of AgNPs and AuNPs from secretions of living organisms that are not killed but returned to nature, thus preserving biodiversity [17,18].

Also, the mucus from different snail species studied shows a wide variety of active substances, such as those collected from the feet of two land snails, *Lissachatina (Achatina) fulica* and *Hemiplecta distincta*, which have different composition, properties, and antimicrobial activity [19]. The published results of phenolic compounds in the beetle’s guard gland extract showed that they could act as a stabilizing agent for the synthesis of AgNPs and also exhibited anticancer activity and cytotoxicity against a Dalton’s lymphoma (DLA) cell line [20]. Also, ecofriendly synthesized biogenic AgNPs-SM from the mucus of *Achatina fulica* snail by Mane et al. showed strong antibacterial, antifungal, as well as anticancer activity against Hela cells [21]. Of particular interest is the potential of *Achatina fulica* mucilage in the green synthesis of bio-nanocomposites of copper oxide (SM-CuONC) and cobalt oxide (SM-Co_3_O_4_NC), with highly pronounced anticancer activity against human breast cancer (MDA-MB-231), colon cancer (HCT-15), cervical cancer (HeLa), etc. Both bionanocomposites showed better antioxidant activity and 100% larvicidal activity against mosquitoes [22].

Snails are a large and diverse group of organisms belonging to the Gastropod, a class of the phylum Mollusca. One of the most common land snail species is the brown garden snail *Cornu aspersum*, which is a rich source of biologically active natural substances such as peptides, proteins, glycoproteins, and hyaluronic and glycolic acid. They have long been documented to be of importance to health benefits, which is why they are incorporated into drugs to treat various human diseases [23]. For example, the chemical composition and mineral profile of *Helix aspersa* Müller mucus are related to its antimicrobial properties and its importance in wound healing [24,25,26,27]. Several results have been published on the toxicity of *H. aspersa* mucus extracts against eukaryotic cells. Trapella et al. reported a lack of toxicity and a proven cytostatic effect at all concentrations tested in vitro of mucus extracts (HelixComplex) on human dermal fibroblasts (MRC-5) [28]. This was also confirmed by a study by Gentili et al. on the non-toxic effect of HelixComplex on human keratinocytes [29].

Mucus has been found to protect cells from apoptosis and to significantly induce cell proliferation and migration through direct and indirect mechanisms. Research by Deng et al. showed strong adhesion to wet tissue of mucus extracts from two land snails, hemostatic effect, good biocompatibility, hemocompatibility, and in vivo pro-healing activity for skin wounds [30]. The mucus of the land snail *H. aspersa* was found to play a crucial role as a stabilizing agent in the synthesis of bioactive AgNPs [31]. Purified mucus from the snail *Helix aspersa* allows for the formation of biogenic AgNPs under very mild pH and temperature conditions: at room temperature and at neutral pH. The AgNPs produced inhibited bacterial growth and were used as a reference sample [31]. The synthesis of copper oxide nanoparticles (CuONPs) using *C. aspersum* mucus has not been documented in the literature. Moreover, CuONPs are among the group of oligodynamic noble metals [32] that have received the most recognition for their applications in molecular diagnostics, antimicrobial and antiaging therapies, and medicine [14,32]. 

Based on our previous studies, the present study is aimed at the environmentally friendly synthesis of CuONPs and reducing agents from *C. aspersum* snail mucus, and also at determining the bioactivity of mucus proteins incorporated into copper oxide nanoparticles with mucus components CuONPs-Muc.

## 2. Results

### 2.1. Isolation and Characterization of Mucus Extract from Garden Snail C. aspersum

#### 2.1.1. Isolation of Mucus Extract from Garden Snail *C. aspersum*

This study analyzed an extract of mucus collected from the snails *C. aspersum*, grown in Bulgarian eco-farms, using patented technology [33]. After 24 h dialysis and concentration with a 20 kDa membrane, the snail mucus solution was purified of additives and enriched with proteins with molecular weight (MW) higher than 20 kDa. Comparative analysis of the UV spectra of the starting mucus (Figure 1) and the resulting fraction after dialysis and membrane concentration confirmed the purity of the fraction with predominant proteins with MW > 20 kDa. 

The total concentration of protein in the sample was determined based on the obtained spectrum by measuring the absorbance at 280 nm (Figure 1), but this method is not the most accurate because it is based on the presence of the amino acids tryptophan (Trp), tyrosine (Tyr), and phenylalanine (Phe) in the protein composition. After applying the Bradford method, which is more sensitive than bicinchoninic acid (BCA) protein assay, a total protein concentration of 1.56 mg/mL in the snail mucus obtained was determined.

#### 2.1.2. Characterization of Mucus Extracts from Garden Snail *C. aspersum*

The mucus of *C. aspersum* snails is mainly a mixture of proteins, glycoproteins, acids, etc. Proteins in the fraction with MW > 20 kDa were characterized using 1D-polyacrylamide gel electrophoresis (1D-PAGE) and mass spectrometric analysis (Figure 2A,B). 

The electrophoretic profile of the *C. aspersum* mucus fraction with MW > 20 kDa determined using 12% SDS-PAGE clearly shows eight major protein bands on the gel with MWs around 22, 26, 30, 35, 48, 58, 80, 170, and 250 kDa (Figure 2A). The exact MWs of 22 431, 26 539, 30 520, 35 528, 48 602, and 78 933 Da of the proteins in this fraction were determined from the MS spectrum using matrix-assisted laser desorption/ionization time-of-flight mass spectrometry (MALDI-Tof-MS) (Figure 2B).

### 2.2. Green Synthesis and Characterization of CuONPs-Muc Using Snail Mucus Extract 

#### 2.2.1. Green Synthesis of CuONPs-Muc Using Snail Mucus Extract with MW > 20 kDa

A dialysis-purified and concentrated fraction of *C. aspersum* mucus, mainly rich in proteins with MW > 20 kDa, was used in this study to synthesize CuONPs-Muc and evaluate their antimicrobial activity. The synthesis of CuONPs-Muc depends on various factors, such as the concentration of the starting mucus sample and of the copper ions. The CuONPs-Muc were obtained after mixing 100 mL of snail mucus fraction with MW > 20 kDa with 100 mL of 0.1 M CuSO_4_ added gradually under continuous stirring at room temperature. 

In the synthesis of nanoparticles, the reaction time is also an important parameter, and the synthesis of CuONPs-Muc was followed for 5 days. The visible color change of the CuSO_4_ solution from pale blue to dark green indicated that the formation of CuONPs-Muc started 2 days after mixing, becoming more intense with time until the 3rd day. 

After centrifugation at 4000 rpm and 12 000 rpm for 10 min and removal of excess products, washing three times with water and acetone, and drying at 70 °C for 12 h, the CuONPs obtained were subjected to analytical analyses using UV–vis, high-performance liquid chromatography (HPLC), one-dimensional polyacrylamide gel electrophoresis (1D-PAGE), Fourier transform infrared spectroscopy (FTIR), scanning electron microscopy (SEM), Raman spectroscopy and imaging, and thermogravimetric analysis (TG-DSC).

#### 2.2.2. Characterization of CuONPs-Muc Using UV–Vis and Fluorescence Spectroscopy

The effect of reaction time on the formation of CuONPs-Muc was followed by recording UV–vis spectra at different times of incubation of the reaction mixture of mucus fraction with MW > 20 kDa and copper sulfate. 

The recorded UV–vis spectra after 60 min, 3 h, 6 h, 24 h, 2 days, and 3 days confirmed the formation of CuONPs-Muc after 3 days of incubation by the most pronounced maximum at 456 nm (Figure 3). The formation of CuONPs-Muc was investigated using the exceedingly sensitive method of fluorescence spectroscopy. A comparison of the emission spectra of the mucus fraction with MW > 20 kDa used in the green synthesis and the fraction after obtaining CuONPs-Muc, shown in Figure 4A, after excitation at λ 280 nm reflects a change in intensity (h = 64 mm to h = 40 mm, respectively) and the position of the maximum fluorescence intensity (λ_max_) (from 351 nm to 316 nm, respectively).

A change in the intensity (from h = 87 mm to h = 32 mm, respectively) and position of λ_max_ (from 355 nm to 341 nm, respectively) was also observed after excitation at λ 295 nm for both samples (Figure 4B). These changes resulted from a change in the location of some exposed indole cores of tryptophans from the surface of the proteins to the interior of the CuONPs-Muc formed.

#### 2.2.3. Characterization of the Obtained CuONPs-Muc by Scanning Electron Microscopy Combined with Energy-Dispersive Spectroscopy (SEM/EDS)

SEM/EDS (JEOL JSM6390 and Oxford Instruments, Abingdon, UK) are used to obtain information on the morphology and shape of the CuONPs, as well as their elemental composition and size distribution. 

The shape of the CuONPs-Muc is almost round, spherical; some of them are agglomerated in larger grains; nevertheless, individual particles could be distinguished from one another (Figure 5A). Thus, we were able to evaluate the overall size distribution, which clearly shows that the main sizes of the CuONPs-Muc are in the interval between 75 and 200 nm (Figure 5B).

EDS analysis showed the purity of the synthesized CuONPs-Muc, as copper (Cu) was obtained in a large amount, as shown in Figure 6.

#### 2.2.4. Characterization of CuONPs-Muc Using Raman Spectroscopy and Imaging

As shown in Figure 7, the Raman spectra for the two samples, pure CuO and CuONPs-Muc formed from mucus with MW > 20 kDa with CuSO_4_, were recorded at a 532 nm laser wavelength. The characteristic nanoscale peaks of CuO at 107, 284, and 550 cm^−1^ (Figure 7A) are observed, as well as peaks for the valence vibrations of the S-O bond from the (SO_4_)^−2^ ion, at 1015, 1044, and 1098 cm^−1^ (Figure 7B) [34]. 

#### 2.2.5. Characterization of CuONPs-Muc Using IR Spectroscopy

To identify the proteins in the mucus fraction with MW > 20 kD involved in CuONP-Muc synthesis, FTIR spectra of the starting slime and of the resulting copper system synthesized with CuSO_4_ were recorded (Figure 8). Comparative analysis of the infrared spectra of the starting sample before and after dialysis showed several broad bands between 1453 and 1500 cm^−1^ that were not detected in the dialyzed sample and indicated the removal of additives in the starting mucus after dialysis.

The FTIR spectra presented in Figure 8 show different vibrational bands between 500 and 4000 cm^−1^, which reflect the changes in the functional groups of the initial sample and after CuONP-Muc formation. 

The IR spectrum of the synthesized CuONPs-Muc proves the presence of an organic matrix surrounding the copper core with a broad band with a maximum at 3251 cm^−1^, which reflects the stretching of oxygen–hydrogen (-OH) vibrations (Table 1). The additional band at 2978 cm^−1^ confirms the presence of traces of carbon, reflecting C–H stretching vibrations [35]. 

Two other peaks for the extract and nanoparticles, observed at 1641 and 1530 cm^−1^, are related to C=O bond stretching vibration, which is typical of amides (amide I and amide II). Both bands were much more intense after dialysis, reflecting the increase in protein content of the sample. The absorption at 1641 cm^−1^ associated with amide I reflects the stretching vibrations of the C=O bond of the amide, while the absorption at 1530 cm^−1^ associated with amide II is mainly due to the bending vibrations of the NH bond. 

Other important bands are observed at 661 and 616 cm^−1^ due to the stretching of Cu(I)–O particles, with the vibrational band of CuO at 616 cm^−1^ confirming the presence of Cu nanoparticles. The band at 1414 cm^−1^ in the spectra of both samples reflects the C–N stretching vibrations of the amines and carboxylate moieties associated with the acid derivatives of sugars and amino acid chains. The presence of glycosylated proteins is represented by the broad band centered at 1102 cm^−1^ associated with CO bonds [36].

### 2.3. Characterization of CuONP-Muc Stability Using TG/DSC-MS Analysis

The stability of the synthesized CuONPs-Muc was analyzed using thermogravimetric TG-DSC methods, and the interactions between the proteins from the mucus and copper ions were determined, as well as their thermal stability. The TG curves reflect changes in the weight of the sample depending on the temperature rise, while the processes occurring over time as a result of heating the material are traced by the DTG curves. 

The TG and DSC results are presented in Figure 9 and Table 2 and Table 3, and clearly show a great difference in the thermal degradation processes of pure mucus and CuONPs-Muc. Analyzing the TG-DSC results for both samples, two stages can be distinguished in the heating processes in an argon atmosphere in the range of 25–300 °C. The first stage covers 25 to 120 °C, with the endothermic effect at about 80 °C and the mass loss corresponding to evaporation of the physically adsorbed water (T_max_, IDS = 73.3 °C, M_loss_, IDS = 16.94%). 

The endothermic peak for CuONPs-Muc in the same temperature range is also related to the desorption of physically adsorbed water, but it occurs at a lower temperature and with a greater mass loss compared to pure mucus (Tmax, IDS = 62.2 °C, M_loss_, IDS = 23.47%). In both samples, after the endothermic peak due to the release of water, there follows an exothermic peak, which is much more pronounced in the pure mucus and is located at higher temperatures (Tpeak mucus = 135.8 °C, Tpeak composite = 91.5 °C).

The second stage in the temperature range of 120 to 300 °C is characterized by significant mass loss for pure mucus (M_loss_ = 75.75%) and CuONPs-Muc (M_loss_ = 69.44%) and Tpeak mucus = 183.9 °C, Tpeak composite = 158.9 °C, respectively, which continues up to 180 °C. These changes are related to the initial destruction of the material, with complete destruction of the nanoparticles formed occurring above 180 °C. The processes of degradation of protein molecules, which are endothermic, in the composite are associated with the removal of low molecular weight compounds from the surface of copper-containing particles (van der Waals forces). This is the reason for the less pronounced endothermic peak and the more pronounced mass loss (M_loss_ = 69.44%) of the composite sample up to 130 °C. After the separation of the low molecular weight compounds, complete destruction of the material occurs. 

The output mucus contains low molecular weight and high molecular weight proteins. In high molecular weight proteins, the interactions between the side chains in the interior of the molecule are more pronounced, which is related to higher conformational stability compared to low molecular weight proteins. Therefore, the larger enthalpy values (ΔH) determined from the peak area of the DCS curves for the pure mucus sample (ΔH = −1216 J/g) reflect stronger intramolecular interactions in the protein (Table 3).

### 2.4. Identification of Proteins Involved in Green Synthesis of CuONPs

To determine the involved proteins from the mucus in the formation of CuONPs-Muc, two comparative analyses were carried out using reversed-phase high-performance liquid chromatography (RP-HPLC) and one-dimensional polyacrylamide gel electrophoresis (1D-PAGE), comparing the changes in the proteins of the mucus fraction with MW > 20 kDa and of the supernatant after CuONP-Muc preparation and precipitation.

#### 2.4.1. 1D-PAGE and RP-HPLC Analyses

The first analysis was carried out using 12% SDS-PAGE, and Figure 10B presents an electrophoretic probe of the starting fraction used for green synthesis (Figure 10A, line 2) and the supernatant after separation of the synthesized CuONPs-Muc. The original *C. aspersum* mucus sample (Figure 10A, lane 2) shows proteins with MW ~210 kDa, ~100 kDa, and ~60 kDa, which are well represented, as well as with MW between 55–65 kDa and 25–41 kDa.

A change in the two electrophoretic paths shows a decrease in the intensity of the band at 30 kDa and 26 kDa, which are very intense in the starting fraction, as well as a greatly increased intensity of the bands with MW 235 kDa and 160 kDa, presented in Figure 10A, line 3 (synthesized CuONPs). No or very weak change was observed for the proteins with MW ~40 kDa and ~60 kDa. Additional information on the involvement of proteins as a reducing agent is the comparative analysis of the changes in the chromatograms of the eluted proteins on a C18 column (8.00 × 250 mm) of RP-HPLC in the dialyzed mucus with MW > 20 kDa and the fraction after obtaining NPs. The comparative analysis presented in Figure 10B shows a significant change in peaks 3, 5, 7, and 9, which do not elute after NP synthesis. We can assume that these proteins are involved in the formation of CuONPs-Muc. Also, two new peaks (1 and 2) appear, which are probably related to the formation of CuONPs-Muc.

#### 2.4.2. Image Analysis of 10% SDS-PAGE Using ImageQuant™ TL v8.2.0 Software

The identification of proteins was carried out using 10% SDS-PAGE (Figure 11), and the exact MWs of the isolated fractions eluted as peaks 7 and 9 (Figure 10, lanes 2 and 3) were determined.

The electrophoretic profile showed the highest expression of proteins with MW mainly above 100 kDa, as well as decreased expression of proteins with MW 16.715 kDa, 22.474, 51.410 kDa, 65.132 kDa, 78.939 kDa, and 88.913 kDa.

The electrophoretic profile of fractions 7 and 9 eluted using RP-HPLC on a C18 column expressed two major bands with MW 30.59 and 26.55 kDa (positions 2 and 3 on 10% SDS-PAGE, Figure 11). In addition to these proteins (common to both fractions), the electrophoretic analysis of fraction 9 also included some poorly expressed proteins. 

The electrophoretic profile of fractions 7 and 9, isolated on a C18 column, expressed two main bands with MW 30.59 and 26.55 kDa. This is a confirmation of the result presented in Figure 10, where the difference between the supernatant of the *C. aspersa* mucus fraction with MW > 20 kDa is presented before and after CuONP-Muc precipitation.

### 2.5. Antibacterial Activity of CuONPs 

Studying the antibacterial activities of pure mucus and synthesized CuONPs-Muc against the eight Gram-positive and Gram-negative pathogenic bacterial strains revealed that mucus itself has no or very weak inhibitory effect against the tested microorganisms (Figure 12). 

Yet, its use as a reducing agent for synthesizing CuONPs-Muc significantly increased the microbial growth inhibition properties. CuONPs-Muc obtained by *C. aspersum* mucus fraction with Mm > 20 kD had noticeable antimicrobial effects on most of the tested Gram-positive and Gram-negative bacterial strains (Figure 13). However, the observed effect of CuONPs-Muc was significantly lower for Gram-negative bacteria. *E. coli* and *S. typhimurium* showed resistance to the CuONPs-Muc obtained, and no zones of growth inhibition were detected. The other *Salmonella* species, *S. enteritidis*, together with *Stenotrophomonas maltophilia*, showed lower susceptibility to the tested nanomaterials *(*Figure 13A). 

Much better antibacterial activity of CuONPs-Muc was found against Gram-positive test strains, and the measured growth inhibition zones varied between 19 ± 0.9 and 38 ± 1.6 mm (Figure 13B). The highest antibacterial potential of the CuONPs-Muc obtained was detected against both *Bacillus* species at 35 ± 1.2 mm and 38 ± 1.6 mm, respectively. Among all the Gram-positive bacteria tested, *Staphylococcus aureus* showed lower resistance to CuONPs-Muc; its growth was relatively slightly inhibited and the measured zones were like those of the Gram-negative bacteria (10 ± 0.4–19 ± 0.3 mm).

Next, the antibacterial effect of CuONPs-Muc was assessed compared to commonly used commercial antibiotics. The antibiograms performed revealed that the most effective against Gram-positive bacteria were vancomycin (VA) and erythromycin (E), with a growth inhibition zone varying from 12 ± 0.6 to 27 ± 1.0 mm (Figure 14A). At the same time, the CuONPs synthesized from the mucus fraction with MW > 20 kDa showed zones 30% bigger than those of the vancomycin (Figure 14A).

Similar results were observed in Gram-negative bacteria. The efficiency of the CuONPs-Muc against *Salmonella enteritidis* and *Stenotrophomonas maltophilia* is similar to those found for vancomycin, the most effective antibiotic against Gram-negative bacterial strains tested (Figure 14B).

## 3. Discussion

Nanoparticles occupy an increasingly important place in medical therapies as they increase the efficacy of drugs through their proper targeting and distribution in tissues [37,38,39]. Environmentally friendly technologies were used to generate nanoparticles of silver, zinc oxide, and other materials. A suitable source for green synthesis of NPs is the mucus from garden snails of the species “Helix”, which is rich in bioactive substances with potential applications in the prevention and/or treatment of some human diseases.

Various mucus extraction techniques and methods have been used and patented, such as acidic or neutral stimulating solutions, which have been shown to affect the final composition of snail mucus [19,40,41]. To protect the substances present in the mucus, we used extraction without stimulants using our patented method [33]. The UV–vis spectra presented in Figure 1 of snail mucus before and after dialysis confirm the purity of the sample from impurities.

The mucus of land snails has been shown to contain between 75 and 95% water and a mixture of proteoglycans, glycosaminoglycans, glycoprotein enzymes, hyaluronic acid, proteins, antimicrobial peptides, amino acids, and metal ions [42]. However, based on information reported by Xie et al., for the ineffective participation of proteins with MW below 7 kDa in the synthesis of AgNPs [42], the mucus used for CuONPs was enriched in proteins with MW > 20 kDa after concentration on a 20 kDa membrane.

The distribution of proteins in the mucus fraction from snail *C. aspersum* with MW > 20 kDa demonstrated using electrophoretic profiling (Figure 2A) as well as MALDI analysis showed proteins with MWs of 22 431, 26 539, 30 520, 35 528, 48 602, and 78 933 kDa (Figure 2B). The results obtained are consistent with published information on *Actinia equina* mucus, which contains about 24.2% proteins with different MWs of 3.5, 14.3, 20.1, 29.0, 43.0, 66.0, and 97.4 kDa, as well as for mucus proteins from *Mastacembelus armatus* at 34 kDa, 45 kDa, and 144 kDa [43]. It was found that proteins with MW above 100 kDa in the mucus of *H. aspersa* and *H. pomatia* corresponded to glycoproteins [44].

Another published information related to the advantage of using the whole plant extract was used in CuONP synthesis, where all compounds in the extract participate in the process as reducing agents and also as capping agents in complex systems. If only individual isolated compounds are used in the process of NP synthesis, nanoparticles with a smaller size, shape, and uniformity are formed than when using the whole extract. Another important factor is the neutral pH of the medium, which is consistent with the behavior of some biomolecules in mucus, which may be inactive under acidic conditions and will not participate in nanoparticle formation. Based on all these factors, the applied efficient, simple, biogenic, and sustainable synthetic route conducted at room temperature showed CuONP-Muc formation after 5 days of incubation of a mixture of MW > 20 kDa fraction of garden snail mucus with 0.1 M solution of CuSO_4_.

Evidence for the formation of copper nanoparticles and presentation of their physicochemical properties are the data obtained from the various methods and techniques applied, such as UV–vis, HPLC, 1D-PAGE, FTIR, SEM, Raman spectroscopy and imaging, and DTG. The green synthesis of CuONPs of the fraction with MW > 20 kDa was proven using the most pronounced maximum at 456 nm (Figure 3) of UV–vis spectra after 3 days of incubation, as well as the much more sensitive method of fluorescence spectroscopy.

It is known that amino acids, such as proline, tryptophan, or tyrosine, are often used as ligands in nanoparticle preparation processes [45]. Due to their ability to interact with hydrophobic regions in protein secondary structures, they disrupt the natural tendency of selected proteins to form oligomers by stacking β-sheet structures [46]. Aggregation and fibrillation of proteins controlled by nanomaterials can contribute to a change in the behavior of proteins, which is expressed in a change in fluorescence intensity or a shift in the fluorescence emission from some NPs as a result of interaction with proteins [47].

The observed change in the fluorescence analysis of the mucus fraction with MW > 20 kDa before and after the formation of CuONPs-Muc is due to the formation of a protein corona, which leads to a change in the conformation and environment of the tryptophan residues in the proteins. 

The fluorescent amino acid Trp, Tyr, and Phe residues of the protein are indicators of the changes in the physical and chemical properties of the environment in which the fluorophore is located in the realized conformation. They reflect the changes that take place both in the environment and inside the protein molecule itself. These changes are expressed in the fluorescence spectra with a shift in the maximum and emission intensity [48]. Protein absorption at 280 nm is known to be mainly associated with Tyr and Trp residues, while at wavelengths greater than 295 nm, they mainly reflect the absorption of Trp residues [48].

Tryptophan residues located on the surface of proteins in the mucus fraction with MW > 20 kDa are responsible for the fluorescence emission at 351 nm (λ_ex_ = 295 nm). After the formation of biogenic nanoparticles, they find themselves in a more hydrophobic environment, buried in the protein corona of the CuONPs-Muc formed. As a result, a blue shift of the fluorescence emission at 341 nm (λ_ex_ = 295 nm) and a significant decrease in its intensity were observed. The present fluorescence study shows the reorganization of the secondary structure and spatial conformation (or three-dimensional conformation) of proteins in the mucus fraction with MW > 20 kDa, supporting the formation of CuONPs-Muc as reducing and stabilizing agents.

Comparative analysis of the emission spectra of the initial sample with MW > 20 kDa and after CuONP-Muc synthesis was performed under excitation at 280 nm and 295 nm. The emission spectra presented in Figure 2, recorded after excitation at 280 nm, reflect a change in intensity (from 64 mm to 40 mm, respectively) and the position of the fluorescence emission maximum, λ max (from 351 nm to 316 nm, respectively). This change can be explained by the location of the Tyr and Thr residues in the molecule, as λ_max_ = 351 nm reflects the fluorophores exposed on the surface of the proteins and access to them. The fluorescence intensity decreases in the sample after the formation of CuONPs-Muc, which is the result of the coagulation of the proteins in nanoparticles, as a result of which a part of the fluorophores are buried in the CuONPs-Muc thus formed. This is also confirmed by the shift of λ_max_ = 351 nm to 316 nm, which corresponds to the emission of “buried” fluorophores.

The stated assumption is also supported by the results presented in Figure 4B: emission spectra of both samples after excitation at 295 nm. According to the two-state hypothesis, for Trps located on the surface of the molecule, a change in intensity is observed (from 87 mm to 32 mm, respectively), which can be explained by the role of Trps and, more precisely, surface-localized Trp residues. Also, a shift in the wavelength of the fluorescence emission in the long-wave range from λ_max_ 355 nm to λ_max_ 341 nm was recorded, which is a sign of a realized structural conformation of the proteins. As a result of CuONP-Muc formation, the exposed indole rings of Trps from the surface of proteins are “buried” in the CuONPs-Muc formed, and only a fraction of the indole rings remain accessible, which are exposed on the surface of CuONPs.

SEM analysis (Figure 5) and Raman spectroscopy with peaks at 107, 284, and 550 cm^−1^ (Figure 7) confirm the formation of CuONPs-Muc. The images reveal that the CuONPs are agglomerated and form larger grains (Figure 5) with a size in the range of 100–150 nm. The high concentration of Cu, O, C, and S presented by the EDS analysis indicates the purity of the synthesized CuONPs-Muc, as well as the presence of unreacted copper ions. This result was also confirmed by the Raman spectra recorded at a wavelength of 532 nm, as peaks for valence vibrations of the S-O bond of (SO_4_)^−2^ ion were also observed at 1015, 1044, and 1098 cm^−1^.

The formation of NPs is mainly the result of two types of interactions: between Cu ions and the side groups of proteins, as well as between the outer surfaces of the mucus proteins themselves. The formation of NPs is based on intermolecular forces such as van der Waals, electrostatic, covalent, hydrogen bonds, and π–π arrangement [45]. The crucial importance of proteins and enzymes for the creation and stability of nanoparticles has been demonstrated by FT-IR analysis of fungal biomass filtrate [49]. Snail mucus contains a wide variety of active compounds, such as allantoin, collagen, elastin, natural peptides, proteins, and enzymes (superoxide dismutase—SOD and glutathione S-transferase—GST); polyphenols; glycolic, hyaluronic, and lactic acid; vitamins A, C, and E; as well as metal ions—copper (Cu), iron (Fe), and zinc (Zn) [50]. The involvement of some of these proteins and enzymes in the green synthesis of CuONPs from the mucus of the garden snail *C. aspersa* was demonstrated by the changes in the functional groups on the surface of the CuONPs-Muc. The strong absorption peaks at 3251 cm^−1^, 2924 cm^−1^, 1641 cm^−1^, 661 cm^−1^, and 620 cm^−1^ show similar behavior with the different functional groups in the synthesized nanoparticles, such as OH groups of phenols and alcohols, the NH groups of amino acids in proteins, carboxyl (COO) residues, etc. 

The observed IR spectra of amide I are associated with the C-O bond vibrations of α-helical (1630–1655 cm^−1^), β-sheet (1660–1690 cm^−1^), and disordered (1660–1670 cm^−1^) structures that are used to study the secondary structure of proteins. The deformation N-H and valence C-N bonds of the peptide group of amide II are in the region of 1520–1580 cm^−1^. The change in the absorption bands at 1641 cm^−1^ and 1530 cm^−1^—well separated in the infrared spectrum of dialyzed mucus but, after the formation of CuONPs, merging and decreasing in intensity—confirms the involvement of glycoproteins in the synthesis of CuONPs. Another important evidence for the involvement of glycoproteins in the synthesized NPs is expressed by a decrease in the intensity of the peak at 1414 cm^−1^, which reflects the carboxylate moieties associated with the acid derivatives of sugars and amino acid chains, as well as an increase in the intensity of the peak at 1102 cm^−1^, related to CO bonds.

Supplementing the information on the involvement of proteins in agglomerates is TG/DSC-MS analysis, which reflects a difference in the stability of the forms. A lower stability of CuONPs-Muc in biological solutions (i.e., phosphate buffered saline) [51,52] and a greater thermal stability of the composite have been reported, which are associated with the adsorption of more high molecular weight compounds from the protein [53]. 

The information presented in Figure 9 and Table 1 and Table 2, on the interactions between the protein and CuONPs-Muc, as well as its thermal stability, from TG-DSC analyses, shows a significant difference in the behavior of the two samples. An explanation for the faster decomposition of the composite compared to the pure mucus may be due to the mainly low molecular weight compounds adsorbed from the mucus that form the CuONPs-Muc. Moreover, the larger enthalpy values (Δ*H*) determined from the peak area of the DSC curves for the pure mucus sample (Δ*H* = −1216 J/g) reflect more favorable intramolecular interactions in the protein.

This difference can be explained by the proteins in the mucus having different molecular weights. Protein size mainly affects the conformational change, folding and unfolding of proteins, and also the surface coverage of NPs. Proteins can be divided into three subcategories: small “hard” proteins with a rigid structure, large “soft” proteins with a flexible structure, and medium proteins. Small proteins do not undergo conformational changes upon contact with the substrate surface, whereas larger proteins undergo greater conformational changes, caused by their contact with the NP surface [54]. Therefore, the lower thermodynamic affinity of “stiff” proteins than that of larger proteins can be explained by the weaker conformational changes in the structure, reflected in a smaller number of binding points per protein molecule [55].

More precise information about the proteins that are involved in the formation of the NPs is presented using RP-HPLC analysis and 1D-PAGE, comparing the changes in the proteins of the mucus with MW > 20 kDa and of the supernatant, after obtaining and precipitation of CuONPs-Muc. The changes observed in peaks 3, 5, 7, and 9 were confirmed electrophoretically (Figure 10B) and reflect the proteins involved in the formation of CuONPs-Muc. These peaks were associated with proteins with MW of about 30 and 26 kDa on lane 2 at 12% SDS-PAGE, which were not observed on lane 3, suggesting the involvement of proteins as a reducing agent in CuONPs (Figure 10). Also, new peaks 1 and 2 appeared, which were probably related to the formation of CuONPs-Muc.

Additional information is provided by ImageQuant^TM^ TL analysis of the proteins with MW 30.691 kDa and 26.549 kDa, which were incorporated into CuONPs-Muc and eluted as peaks 7 and 9. Based on published data on the proteins contained in *C. aspersum* mucus, we can assume that the proteins with MW 26 549 kDa and 30 591 kDa most likely exhibit antioxidant activity and belong to the families of superoxide dismutase, glutathione peroxidase (GPx), and glutathione transferases (GST). Brieva et al. found that the mucus of *Cryptomphalus aspersa* (also known as *H. aspersa*) contains antioxidant superoxide dismutase and glutathione transferase activity [56]. 

Several studies in recent years have also confirmed the antioxidant properties of a *C. aspersum* mucus fraction with MW 10–30 kDa and a fraction with MW > 20 kDa [57,58,59].

Proteins from mollusks and gastropods are represented in the UniProt Knowledgebase (UniProtKB) as superoxide dismutase subunit [Cu-Zn]-like protein from *Pomacea canaliculata* with MW 15.6 kDa (XP_025098994) and 15.774 kDa from *Lymnaea stagnalis* (Q7YXL9) and glutathione S-transferases with MW 27.978 kDa (from *P. canaliculata*, A0A2T7PWN7); with MW 27.774 kDa (from *Haliotis rubra*, XM_025244070.1). The results obtained for a protein with an MW of 26.549 kDa agreed with an enzyme of the glutathione S-transferases family with an MW between 20.5 and 27.5 kDa.

The protein with MW 30.59 kDa is in good agreement with lectins and is also related to the antimicrobial properties of mucus [60]. The protein detected at 37.52 kDa is a major protein identified in *C. aspersum* mucus, with MW 37.4 kDa (QEG59312.1) [60]. Based on the presented results, we hypothesize that mucus proteins with MW 26.55 kDa and 30.59 kDa play a major role in forming CuONPs-Muc.

Many reports of antibacterial, antifungal, larvicidal, anti-termite [61,62,63,64,65], and anticancer [66,67] activities of CuONPs derived from plants, animals, and microorganisms. Furthermore, several studies conducted in recent years have demonstrated the advantages of employing snail mucus to synthesize Ag and AuNPs, which have been shown to have broad-spectrum antibacterial activity, anticancer potential, and anti-inflammatory properties [21,68]. Based on the presented investigations, it can be assumed that CuONPs obtained from the *C. aspersum* mucus with MW > 20 kDa also show in vitro antibacterial activity. However, higher susceptibility was detected for Gram-positive species, which data corresponds to other authors’ results [69,70]. This is probably due to the specificity in cell wall chemical composition and structure of Gram-positive bacterial cells—stronger negative surface charge, allowing positive NPs to attract, and the presence of teichoic acid, whose phosphate chains interact with NPs, thus preventing their aggregation. The presence of a high number of pores on Gram+ cell walls additionally allows penetration of NPs into the cells and causes cell death [60,71]. 

The detected efficacy of the synthesized CuONPs-Muc is lower against Gram-negative bacterial strains. However, these findings are similar to the studies of other authors who revealed that the minimum inhibitory concentrations (MICs) of ultra-small CuO nanoparticles against *E. coli* is 40% higher than those for *S. aureus* [72]. It is assumed that the key factors responsible for bacterial growth inhibition by CuONPs are the electrostatic interactions and cell wall adherence. As a consequence, ROS are produced due to the Cu^2+^ dissociation. These ions have the ability to infiltrate cells and break their membranes, which can lead to bacterial cell leakage and a disturbance in the interior content of the cell [73]. Considering that the envelope of Gram-negative bacterial cells consists of both inner and outer membrane, it could be assumed that higher concentrations of CuONPs are required to hinder cellular growth.

Surprisingly, the efficacy of the tested CuONPs-Muc was similar to or even higher than that of commonly used antimicrobial drugs. Antibiotics such as vancomycin and erythromycin are often used to treat infections brought on by Gram-positive bacteria that are resistant to many medications, particularly methicillin-resistant *Staphylococcus aureus* [74,75]. Vancomycin has been utilized extensively in hospital settings during the past few decades as a result of the constantly increasing number of multidrug-resistant strains each year [72]. To address these serious problems in the postantibiotic era, new strategies for the treatment of microbial infections are urgently needed. One such new type of antibacterial agent is nanomaterials. 

The antibacterial effect of metal nanoparticles is based on a set of their specific properties, such as the ability to destroy or prevent the formation of microbial biofilms, generation of reactive oxygen species (ROS) that can damage DNA, RNA, and proteins, inhibition of replication by binding to DNA, etc., thus killing microorganisms in ways against which the emergence of resistance is limited [69,73]. Therefore, the extensive inhibitory antibacterial potential exhibited by biosynthesized CuONPs using snail mucus makes them promising preparations for future medical applications, such as alternatives to antiseptics and antibiotics.

Our research will continue the topic of obtaining biogenic metal nanoparticles (from copper, zinc, and silver ions) in the mucus matrix of the garden snail *C. aspersum* under different conditions, including in the presence of ascorbic acid, as well as their characterization and biomedical applications. In order to clarify the mechanism of antimicrobial activity of the new biogenic nanoparticles obtained, the cellular redox status, the level of transcription, and the levels of apoptotic cell markers in the tested pathogenic microorganisms will be monitored before and after inoculation with novel NPs-Muc. Combining proteomics, mass spectrometry, and bioinformatics, the changes in the proteome maps of bacterial and fungal cells before and after treatment with the most active NPs-Muc will be tracked and the key proteins responsible for the relevant biological activities will be identified. In this way, our studies will lead to obtaining important new information that will serve to develop a new generation of antimicrobial therapeutics as an alternative to conventional antibiotic treatment and drug multiresistance.

## 4. Materials and Methods

### 4.1. Sample Preparation

The crude snail mucus was obtained from certified environmentally clean snail farms on the territory of Bulgaria using a patented method of extraction and filtration of mucus from the garden snails *C*. *aspersa,* so the snails survived without disturbing their biological functions [33]. After several steps of filtration, the snail mucus solution was freed of chloride ions and additives after dialysis for 24 h. The crude mucus extract was separated into two fractions after ultrafiltration using a membrane system (ӒKTA pure™) with a column at 20 kDa. A peptide fraction with MW above 20 kDa was used for the synthesis of CuO-NPs-Muc from snail mucus. The protein concentration was determined using the Bradford method [76]. 

### 4.2. 1D Polyacrylamide Gel Electrophoresis (1D-PAGE)

Protein fractions were analyzed by 12% and 10% sodium dodecyl sulfate-polyacrylamide gel electrophoresis (SDS-PAGE), according to the Laemmli method with modifications [77] and visualized by staining with Coomassie Brilliant Blue G-250. Molecular markers of standard proteins with a molecular weight from 6.5 kDa to 200 kDa (SERVA Prestained SDS PAGE Protein Marker 6.5–200 kDa, liquid mix, SERVA Electrophoresis GmbH, Heidelberg, Germany, and SigmaMarker^TM^, Sigma-Aldrich, Saint Louis, MO, USA) were used. Because the commercial Laemmli buffer (Leammli Sample Buffer (2×) for SDS PAGE, SERVA Electrophoresis GmbH, Heidelberg, Germany) does not contain any reduction reagent, 10 mM DTT was added as a reducing sample buffer (concentrations refer to 1× sample buffer). Each sample is mixed with an equal volume of the sample buffer thus prepared, after which the samples are denatured by boiling at 100 °C for 5 min. Equal volumes containing 20 μg/lane of the samples dissolved in Laemmli sample buffer and protein standard mixture were run at a voltage of 145 V using the Bio-Rad feeder (Bio-Rad, Hercules, CA, USA).

### 4.3. MALDI-MS Analysis of Proteins

The molecular masses of proteins in the mucus fraction with MW > 20 kDa from snail *C. aspersum* were measured using an Autoflex™III, High-Performance MALDI-TOF & TOF/TOF System (Bruker Daltonics, Billerica, MA, USA), which uses a 200 Hz frequency-tripled Nd–YAG laser operating at a wavelength of 355 nm. The analysis was carried out using α-cyano-4-hydroxycinnamic acid as a matrix. In total, 2.0 μL of the sample was mixed with 2.0 μL of matrix solution (7 mg/mL of α-cyano-4-hydroxycinnamic acid (CHCA) in 50% CN containing 0.1% TFA), and only 1.0 μL of the mixture was spotted on a 192-well stainless steel target plate. They were dried at room temperature and subjected to mass analysis. A total of 3500 shots were acquired in the MS mode, and collision energy of 4200 was applied. The mass spectrometer was externally calibrated with a mixture of angiotensin I, Glu-fibrinopeptide B, ACTH (1–17), and ACTH.

### 4.4. The Synthesis of CuONPs-Muc 

One gram of lyophilized mucus fraction with MW > 20 kDa was dissolved in 40 mL of distilled water (a concentration of 25 mg/mL). A 0.1 M solution of CuSO_4_ was prepared. All solutions were moderately stirred with a magnetic stirrer until complete dissolution (about 30 min). Then, 100 mL of 0,1 M CuSO_4_ solution was added dropwise to 100 mL of snail mucus fraction with MW > 20 kDa under constant stirring at room temperature. The whole mixture, with an initial pH of 5.3, was left in a dark environment with moderate agitation (250 rpm) for 5 days at room temperature (around 20 °C). After the 5-day incubation, samples were centrifuged at 4000 rpm for 10 min. Afterwards, supernatants were separated and centrifuged at 12 000 rpm for 10 min to remove the excess products formed.

### 4.5. Characterization of CuONPs-Muc Using Ultraviolet–Visible Spectroscopy Analysis 

UV–vis absorption measurements were carried out using a UV–vis spectrometer (Shimadzu™ UVmini-1240, Shimadzu Corporation, Kyoto, Japan), using symmetric quartz cuvettes (Quartz Glass High Performance 200 nm–2500 nm, Hellma^®^ absorption cuvettes) with 10 mm optical length at room temperature (25 °C) in the range of 250–500 nm. All absorption spectra were corrected by subtracting the absorption spectrum of the buffer solution in the same wavelength range. During the 5-day incubation, the UV–vis spectrum of the samples was measured at 24 h, 48 h, and 72 h to follow the dynamics of nanoparticle formation. Additionally, UV–vis spectra were also measured after the centrifugations at 4000 rpm and 12 000 rpm (VWR^®^ Micro Star 30R Microcentrifuge, VWR International, Tokyo, Japan). 

### 4.6. Characterization of CuONPs-Muc Using Fluorescence Spectroscopy

Fluorescence studies of samples containing a mucus fraction with MW > 20 kDa before and after the CuONPs-Muc were obtained by green synthesis were performed using a Shimadzu RF-6000 spectrofluorometer (Shimadzu Corporation, Kyoto, Japan) using a xenon lamp as the excitation source and a fluorescent cuvette with a path length of 10 × 10 mm (Hellma 101-QS, Floressan Quartz Küvet), at band ex. = 3 nm and band em. = 20 nm. Fluorescence emission spectra were measured after excitation at λ_ex_ 280 nm and at λ_ex_ 295 nm. For each excitation wavelength, the emission was recorded between 220 and 700 nm, with a step increment of 0.5 nm. The changes observed in the emission spectra of a sample containing a mucus fraction with MW > 20 kDa and a sample containing the CuNPs obtained by green synthesis were compared.

### 4.7. Characterization of CuONPs-Muc Using Raman Spectroscopy and Imaging

Raman spectra were recorded using Raman Microscope Senterra II (Bruker). The intensity of Raman bands is determined by the change in polarizability during normal vibration. Using this technique, it is possible to determine crystal forms, chemical compositions, intermolecular interactions, the degree of ordering, or the spatial distribution of stresses in the tested materials [78]. The samples were placed onto glass (approximately 10 mg) and analyzed using the vertical 20× objective in an 180° backscattering arrangement. The Raman spectrometer parameters used to analyze the CuONPs-Muc include a 532 nm laser wavelength and an exposure time of 100 s; the resolution was 4 cm^−1^ for all samples, and laser power was 6.5 mW.

### 4.8. Characterization of CuONPs-Muc Using FTIR and TG-DSC

Pallets centrifuged multiple times were subjected to FT-IR (Midac MC 2000 Spectrometer), where the FT-IR spectrum was examined in a range of 6000–300 cm^−1^ wavelength interval for the CuONPs-Muc and mucus extract [79].

Thermal analysis (TG) and differential scanning calorimetry (DSC) were performed with a STA 449 Jupiter F3 Netzsch apparatus (Selb, Germany). Two samples were investigated using TG-DSC analysis: a pure mucus sample and composite CuONPs-Muc. The analyses were performed under the following conditions: heating rate 10°K/min in an argon atmosphere with a flow rate of 30 mL/min, temperature range from 25 to 600 °C, and sample weight ~50 mg. A comparison was made between the results obtained from the study of the thermal behavior of the pure mucus sample and CuONPs-Muc in a programmed mode of operation. The ongoing phase transitions in both samples as a result of heating to 600 °C were also compared. Additionally, from the DSC analysis of the mucus and the CuONPs-Muc, the amount of heat Δ*H* released (absorbed) during a given process was calculated for each of the transitions: Tonset, the lowest temperature at which the reaction starts; Tpeak, the maximum transition temperature; Tend, the final temperature of the process. Using TG analysis, the maximum mass loss (Mtotal) was determined for both samples as a result of heating.

### 4.9. Image Analysis of 10% SDS-PAGE by ImageQuant™ TL v8.2.0 Software

After capturing the SDS-PAGE obtained on Image Scanner III (GE Healthcare, Chicago, IL, USA), the image was analyzed using the ‘1D gel analysis’ utility of the Image Quant TL v8.2 software (GE Healthcare Bio-Sciences AB, Uppsala Sweden), which is highly automated software for image analysis. All bands were identified manually using a pen tool, including those in the standard protein marker. To compensate for the intensity of the image background, the background was modified with the “image rectangle” setting. Analysis of the molecular weight of each band was performed using the data for the protein marker in the range 6.5–200 kDa (SigmaMarker^TM^, Sigma-Aldrich, Saint Louis, MO, USA). Automatically, horizontal bands were drawn to the individual bands of the MW marker and calculated using the cubic curve spline. Based on the precalculated number of bands in the marker, the number of bands tested was determined [80].

### 4.10. RP-HPLC Analysis of CuONPs-Muc

Reverse-phase high-performance liquid chromatography (RP-HPLC) on ZURA^®^; HPLC system (KNAUER, Berlin, Germany) was used to detect and identify which proteins of the mucus was capping/reducing the CuO ions. The chromatographic separation was achieved on a Eurospher II 100 C18 (250 × 8 mm) column (KNAUER, Berlin, Germany) by gradient of water–acetonitrile (with 0.1% TFA) for 55 min, which was adjusted as follows: 0–10 min, 100% buffer A (MiliQ-water with 0.1% TFA); 11–15 min, 85% buffer A; 16–40 min, 40% buffer A; 41–45 min, 0% buffer A; 46–47 min, 0% buffer A; 48–49 min, 100% buffer A; 50–55 min, 100% buffer A. The injection volume was 1 mL sample, and the flow rate was 1.5 mL/min at 25 °C. The detection wavelength was 280 nm. 

The samples were centrifuged briefly and filtered through 0.2 µm syringe filters (Minisart NML, hydrophilic, cellulose acetate) before injection. The concentration of the used mucus fraction with MW > 20 kDa was 0.02 g/mL lyophilized sample in Milli-Q water.

### 4.11. Antibacterial Activity of CuONPs

Gram-positive *(B. subtilis* NBIMCC2353, *B. spizizenii* ATCC 6633, *S. aureus* ATCC 6538, *L. innocua* NBIMCC8755) and Gram-negative strains (*E. coli ATCC8739*, *S. enteritidis* NBIMCC8691, *S. typhimurium* ATCC 14028, *S. maltophilia* ATCC 17666) were selected as representative bacterial models to test the antibacterial properties of the samples. The antibacterial activity of mucus and NP samples was studied by the agar well diffusion method, and after 24 h of cultivation, the zones of cell growth inhibition were measured (d, mm). The results were compared to vancomycin antibacterial activity determined using the agar disk diffusion method (30 µg/disk).

The microbial growth inhibitory effect of CuONPs-Muc, obtained from mucus fraction with MW > 20 kDa, was studied using the agar well diffusion method [81]. As a negative control, a fraction of *C. aspersum* mucus was used. For positive control, the antibacterial activity of different commercially available antibiotics was studied: vancomycin (VA), amoxiclav (AMC), erythromycin (E), and cephalexin (CN)—all with concentrations of 30 µg/disk.

Nutrient agar plates were inoculated with 0.1 mL cell suspension (0.5 McFarland) by spreading it over the entire agar surface. Holes (diameter, 8 mm) were punched in agar with a cork borer, and the wells formed were filled with 0.1 mL of each mucus fraction or CuONPs-Muc sample. Then, the agar plates were stored for 4 h at fridge temperature (for antimicrobial agents’ diffusion) and, after that, were transferred at 37 °C for 24 h. The antimicrobial activities of NPs were evaluated by measuring the zone of growth inhibition (diameter, mm), and the presented values were averaged from three different experiments. Microbial susceptibility to different antibiotics was tested according to the CLSI M39 guidelines [82].

## 5. Conclusions

The present research presents for the first time an environmentally friendly method without toxic waste for synthesizing biogenic CuONPs-Muc from a mucus fraction with MW > 20 kDa of the garden snail *C. aspersum*. The biosynthesized CuONPs-Muc were proven and characterized by applying various modern methods (UV–vis and fluorescence spectroscopy, RP-HPLC analysis, electrophoretic analysis, mass spectrometry, SEM and EDS, FTIR and TG-DSC analyses). The CuONPs-Muc showed higher inhibition of the bacterial growth of various Gram-positive (*B. subtilis* NBIMCC2353, *B. spizizenii* ATCC 6633, *S. aureus* ATCC 6538, *L. innocua* NBIMCC8755) and Gram-negative (*E coli* ATCC8739, *S. enteritidis* NBIMCC8691, *S. typhimurium* ATCC 14028, *S. maltophilia* ATCC 17666) bacteria compared to the initial mucus fraction with MW > 20 kDa.

The results of the in vitro studies show that a combination of new biomolecules with antimicrobial properties from the mucus of the snail *C. aspersum* and an ecological approach—biological synthesis of metal nanoparticles—can lead to new effective antimicrobial therapeutics against high-risk pathogens.

Research in this area will continue to obtain and characterize biogenic metal nanoparticles synthesized from the mucus of the snail *C. aspersum* under different conditions, including in the presence of ascorbic acid, and their biomedical applications. To elucidate the mechanism of the antimicrobial activity of the new biogenic nanoparticles obtained, the cellular redox status, the level of transcription, and the levels of apoptotic cell markers in the tested pathogenic microorganisms will be followed before and after inoculation with new NPs-Muc.

## Figures and Tables

**Figure 1 pharmaceuticals-17-00506-f001:**
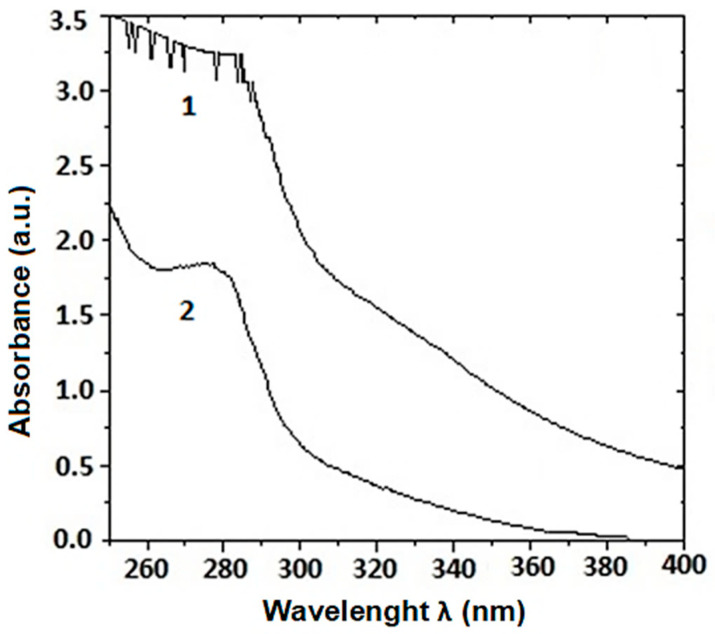
UV–vis spectra of *C. aspersum* mucus with MW > 20 kDa: (**1**) Crude extract; (**2**) Dialyzed and concentrated sample with a 20 kDa membrane.

**Figure 2 pharmaceuticals-17-00506-f002:**
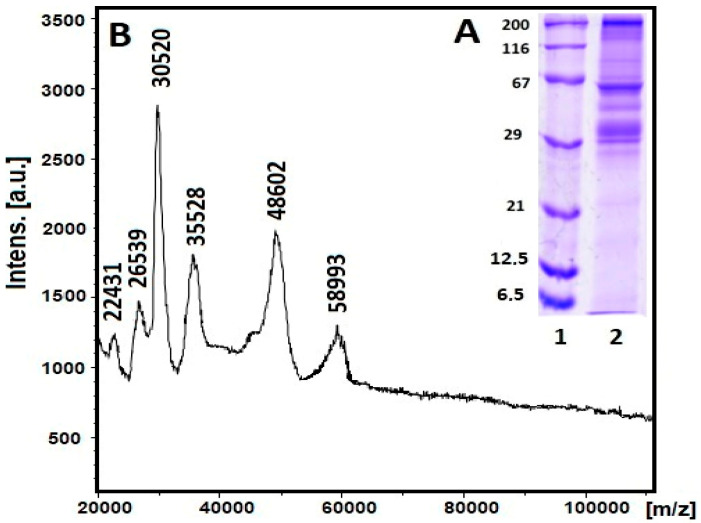
(**A**) 12% SDS–PAGE staining with Coomassie Brilliant Blue G-250: position 1, standard protein marker range 6.5–200 kDa (SERVA Electrophoresis GmbH, Heidelberg, Germany); position 2, extract of *C. aspersum* mucus with MW > 20 kDa; (**B**) MALDI-TOF-MS spectrum of *C. aspersum* mucus, recorded between 20 000 and 100 000 *m/z*.

**Figure 3 pharmaceuticals-17-00506-f003:**
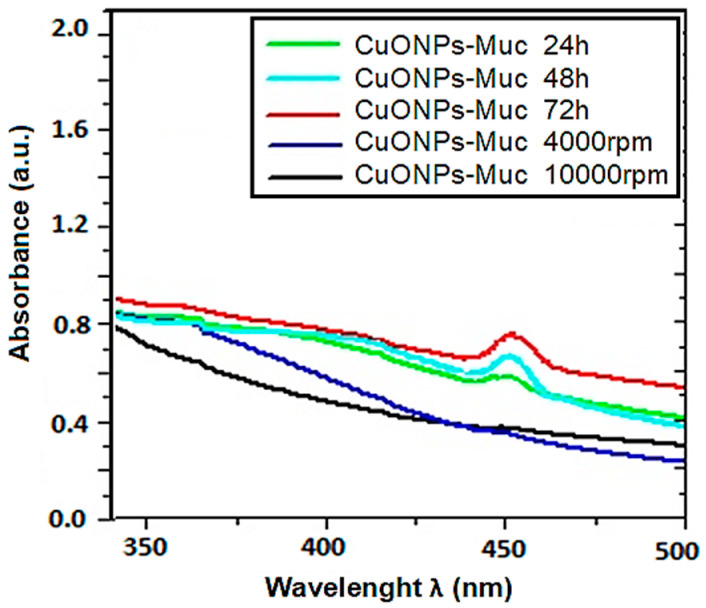
UV–vis spectra of the CuONPs-Muc synthesized at different reaction times. Wavelength absorption curves (Cu_2_SO_4_: extract 1:1, T = 20 °C) after 1 h and centrifugation at 4000 rpm; at 12 000 rpm; after 2 h, 48 h, and 72 h.

**Figure 4 pharmaceuticals-17-00506-f004:**
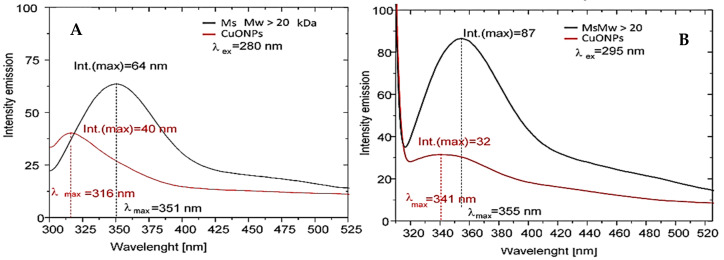
The comparative analysis of the emission spectra of the mucus fraction with MW > 20 kDa used in the green synthesis and the fraction after obtaining CuONPs-Muc (A_280_ = 0.1 in 0.1 M Tris buffer, pH 8). Emission spectra were recorded with a 1 cm quartz cuvette after excitation (**A**) at 280 nm and (**B**) at 295 nm.

**Figure 5 pharmaceuticals-17-00506-f005:**
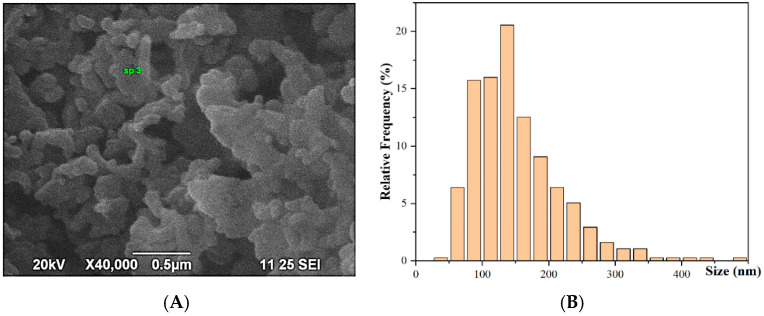
(**A**) SEM images of CuONPs-Muc; (**B**) CuNP size distribution determined from SEM images.

**Figure 6 pharmaceuticals-17-00506-f006:**
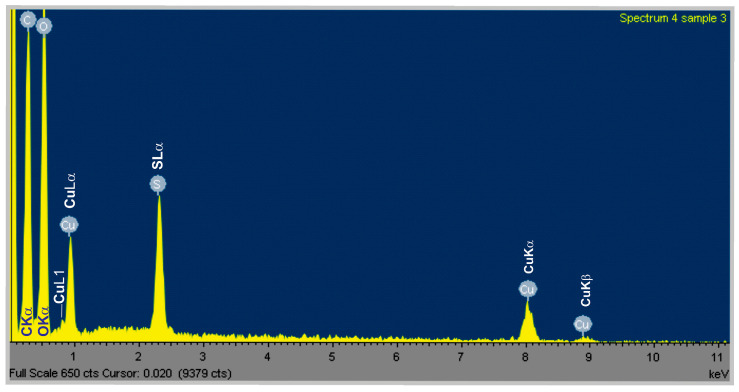
EDS spectra of Cu/Cu_2_O nanoparticles synthesized using snail mucus with MW > 20 kDa.

**Figure 7 pharmaceuticals-17-00506-f007:**
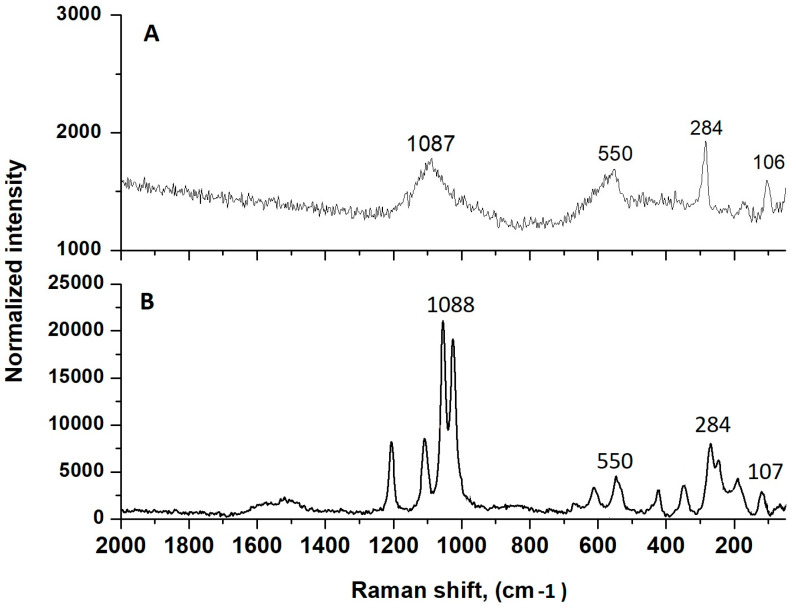
Raman spectra recorded on Raman Microscope Senterra II (Bruker) at a 532 nm laser wavelength for (**A**) pure CuO and (**B**) CuONPs-Muc from mucus with MW > 20 kDa with CuSO_4_.

**Figure 8 pharmaceuticals-17-00506-f008:**
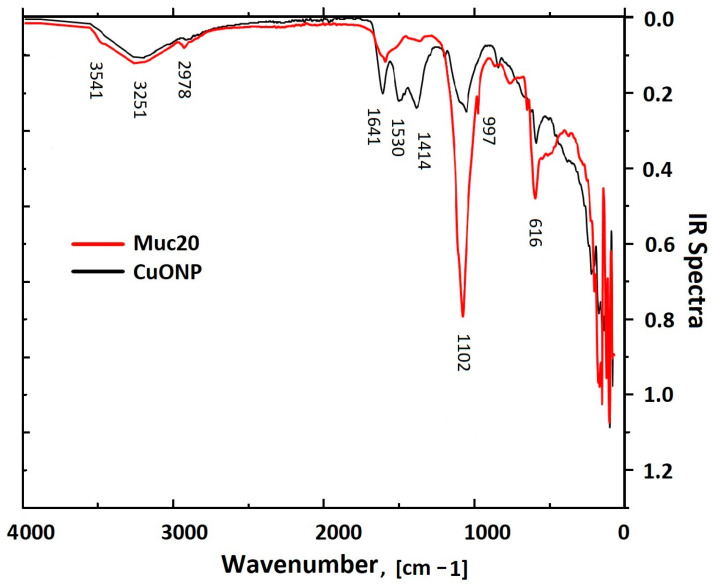
Infrared spectra of dialyzed snail mucus (red line) and dried CuONPs-Muc collected from the synthesis of *C. aspersum* mucus.

**Figure 9 pharmaceuticals-17-00506-f009:**
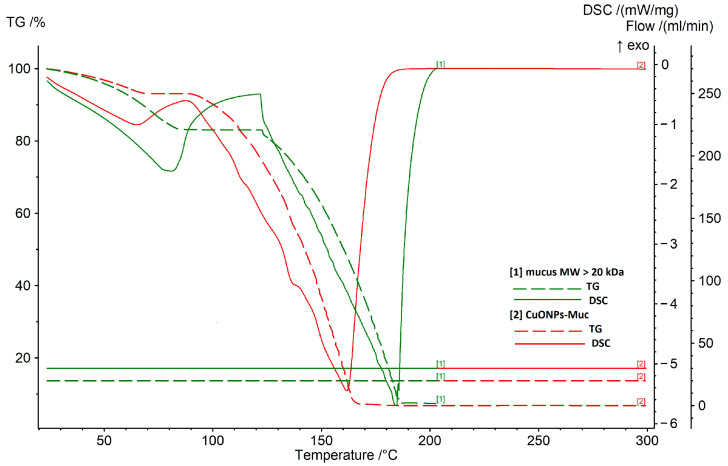
TG-DSC analysis of pure mucus sample with MW > 20 kDa and CuONP-Muc composites.

**Figure 10 pharmaceuticals-17-00506-f010:**
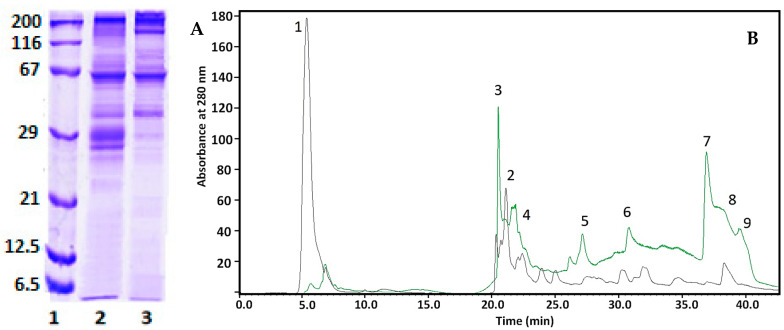
(**A**) 12% SDS-PAGE of line (**1**) standard protein marker in the range 6.5 kDa–200 kDa; line (**2**) mucus fraction with MW > 20 kDa; (**3**) the supernatant after precipitation of CuONPs-Muc. (**B**) HPLC chromatograms of mucus with MW > 20 kDa (black line) and the supernatant after centrifugation of CuONPs-Muc (green line), separated on a C18 column (8.00 × 250 mm).

**Figure 11 pharmaceuticals-17-00506-f011:**
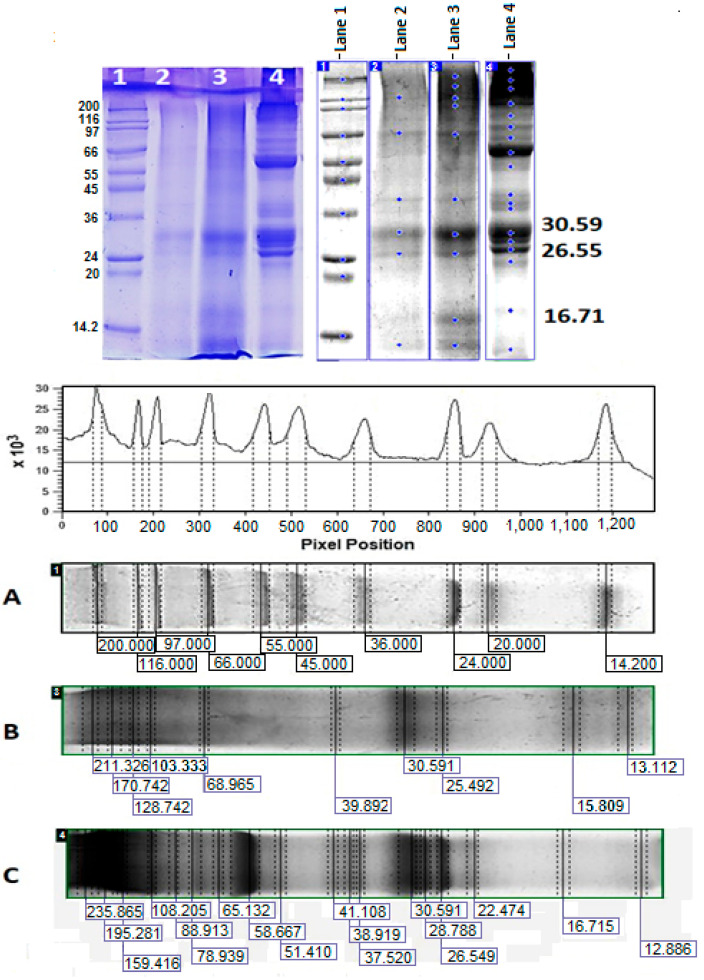
ImageQuant^TM^ TL analysis of 10% SDS-PAGE, visualized with Coomassie G-250 of: (**A**) line 1—standard protein marker in the range 6.5–200 kDa (SigmaMarker^TM^, Sigma-Aldrich, Saint Louis, MO, USA); (**B**) lane 2, peak 7 of the HPLC chromatogram of the supernatant of the *C. aspersa* mucus fraction with MW > 20 kDa after precipitation of CuONPs (Figure 10); (**C**) lane 3—peak 9.

**Figure 12 pharmaceuticals-17-00506-f012:**
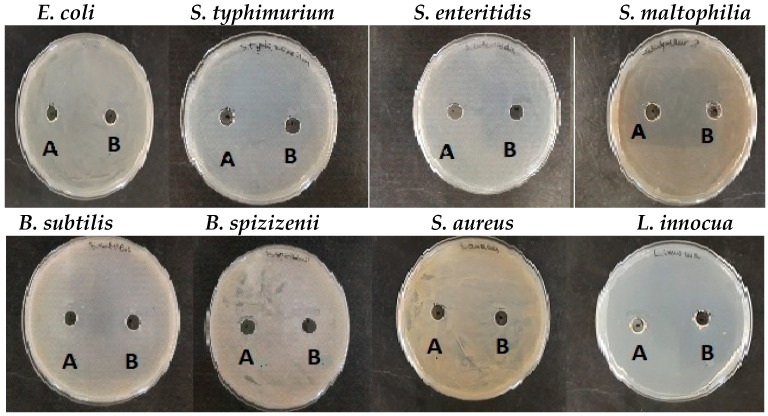
Antibacterial activity of fractions of (**A**) pure mucus from *C. aspersum* with MW < 20 kD and (**B**) fraction with MW > 20 kDa.

**Figure 13 pharmaceuticals-17-00506-f013:**
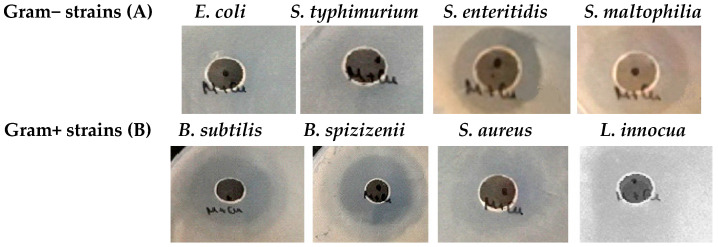
Bacterial growth inhibitory effect of CuONPs-Muc obtained with reducing agent mucus from *C. aspersum* with MW > 20 kDa; antimicrobial activity against (**A**) Gram-negative bacteria; (**B**) Gram-positive bacteria.

**Figure 14 pharmaceuticals-17-00506-f014:**
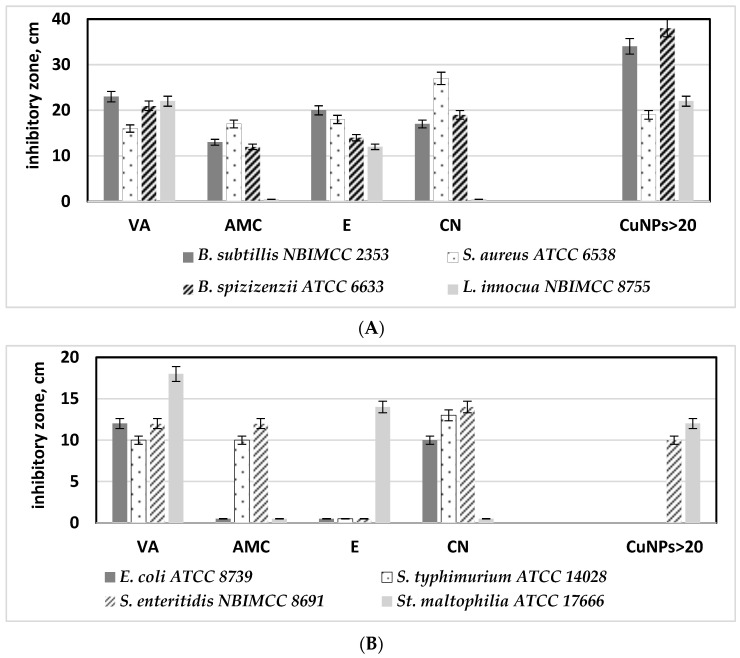
Comparative analysis between the efficacy of commonly used antibiotics and synthetized CuONPs-Muc against tested Gram-positive (**A**) and Gram-negative (**B**) pathogenic bacterial strains: vancomycin (VA), amoxiclav (AMC), erythromycin (E), and cephalexin (CN), CuONPs obtained with a fraction of mucus from *C. aspersum* with MW > 20 kDa (CuONPs > 20).

**Table 1 pharmaceuticals-17-00506-t001:** FT-IR characteristic peaks of the mucus extract of *C. aspersum* and CuONPs-Muc.

Wavenumber [cm^−1^]	Interaction
616	Cu–O
1102	C–O bending
1414	C–N stretching
1530	NH
1641	O–H bending
2978	C–H stretching
3251	O–H stretching

**Table 2 pharmaceuticals-17-00506-t002:** TG –DTG data for pure mucus sample and CuONP composites.

Sample	TG				DTG		
	*M_loss,IDS_* [%]	*M_loss_*_1_ [%]	*M_loss_*_2_ [%]	*M_loss,TOTAL_* [%]	*T_max_*,_*IDS*_ [°C]	*T_max_*_1_ [°C]	*T_max_*_2_ [°C]
	30–120 [°C]	120–200 [°C]	200–300 [°C]				
Mucus	16.94% (9.08 mg)	75.75% (40.64 mg)	0.03% (0.016 mg)	92.72%	73.3	123.0	183.0
CuONPs	23.47% (11.49 mg)	69.44% (34.05 mg)	0.04% (0.019 mg)	92.95%	62.2	134.9	160.9

*M_loss,IDS_*, mass loss in initial decomposition step; *M_loss_*, mass loss in main decomposition step; *M_loss,TOTAL_*, total mass loss; *T_max_,_IDS_*, maximum temperature in initial decomposition step; *T_max__1,2_*, temperature at the maximum degradation step.

**Table 3 pharmaceuticals-17-00506-t003:** DSC data for pure mucus sample and CuONP composites.

Sample	DSC							
	*T_onset_* [°C]	*T_peak_* [°C]	*T_end_* [°C]	∆*H* J/g	*T_onset_* [°C]	*T_peak_* [°C]	*T_end_* [°C]	∆*H* J/g
Mucus	58.2	80.8	91.1	−289.2	127.9	183.9	189.0	−1216
CuONPs	35.3	65.2	85.4	−183.8	125.3	158.9	181.7	−1058

*T_onset_*, temperature of the decomposition initiation; *T_peak_*, peak maximum decomposition temperature; *T_end_*, final decomposition temperature; ∆H, the heat generated during the decomposition reaction obtained by the integration of the thermal peaks (area).

## Data Availability

Data is contained within the article.

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
