# Peer review of "Development of CuO Nanoparticles from the Mucus of Garden Snail Cornu aspersum as New Antimicrobial Agents"

_pharmaceuticals, 2024, doi:10.3390/ph17040506_

Round 1

Reviewer 1 Report

Comments and Suggestions for Authors

The synthesis and study of metal nanoparticles has attracted increasing interest from researchers each year. However, the physical and chemical methods used often have disadvantages associated with the release of toxic chemicals. A solution to this issue is the use of "green" synthesis methods for metal nanoparticles. Biogenic methods are mainly limited to using microorganisms and plant extracts for synthesizing nanoparticles. However, there have been few studies conducted on animal secretions, and even fewer on mollusk secretions. This work focuses on the synthesis and characterization of copper oxide nanoparticles using purified (after 24-hour dialysis and concentration with a 20 kDa membrane) garden snail mucus.

The article needs to increase the evidence base in order to be published in the journal "Pharmaceutics". I have highlighted some comments and suggestions below that might help the authors:

1. How quantitative is the reduction of the copper salt used (CuSO4), how was this controlled?

2. Are there any chemical changes in the concentrated mucus fraction during nanoparticle synthesis? Why is the reduction of Cu ions occurring: what components of the mucus are providing this? What is the nature of the interaction between mucus components and Cu nanoparticles?

3. Considering that the size of nanoparticles is essential for antimicrobial activity, it must be determined accurately. You can clearly determine the dimensions using TEM. From Fig. 5 obtained using SEM, it is not possible to accurately determine the sizes of individual particles. The indicated sizes of 70-200 nm most likely correspond not to copper oxide nanoparticles, but to a complex of nanoparticles with components of the mucus used for the synthesis.

4. In the caption to Fig. 5 A): CuONPs-SS, what does SS mean? Fig. 5 B): The sizes of copper nanoparticles (CuNPs) rather than copper oxide (CuONPs) are analyzed.

5. To avoid confusion, you need to bring the names of the resulting compounds into uniformity. It is more correct to call copper oxide nanoparticles with mucus components CuONPs-Muc.

6. The conclusion that exposed indole tryptophan nuclei from the surface of proteins penetrated into the resulting CuO nanoparticles requires clarification. (167 stock)

8. Requires more detailed explanation - “more favorable intramolecular interactions in the protein” (line 277).

9. The first mention of the latin name of microorganism in the text should be complete, not abbreviated.

10. In the text on line 361, it is stated that vancomycin (V) and erythromycin (E) are most effective against gram-positive bacteria. However, Fig. 14A shows that cephalexin (CN) has greater antibiotic activity against three out of four strains, whereas erythromycin is only effective against one out of four.

11. The abstract, keywords, and conclusions of the article discuss fungal strains Candida albicans ATCC 10231 and Saccharomyces cerevisiae NBIMCC 584, but the text of the paper does not contain information on experiments or their results with these organisms.

Author Response

We would like to thank the reviewers and the Editor for the time and effort taken to improve our manuscript “Development of CuO nanoparticles from mucus of garden snail Cornu aspersum as new antimicrobial agents” Pavlina Dolashka, Karina Marinova, Petar Petrov, Ventsislava Petrova, Bogdan Ranguelov, Stela Atanasova-Vladimirova, Dimitar Kaynarov, Ivanka Stoycheva, Emiliya Pisareva, Anna Tomova, Angelina Kosateva, Lyudmila Velkova, and Alexander Dolashki.

We have revised the manuscript according to all comments. We appreciate the opinion of the reviewers and agree with their comments. The authors have carefully considered the comments and tried our best to address every one of them. We hope the manuscript, after careful revisions, meets your high standards. The revised parts are marked in bold in the revised manuscript.

Comments and Suggestions for Authors and our answers

Reviewer 1

Comments and Suggestions for Authors

The synthesis and study of metal nanoparticles has attracted increasing interest from researchers each year. However, the physical and chemical methods used often have disadvantages associated with the release of toxic chemicals. A solution to this issue is the use of "green" synthesis methods for metal nanoparticles. Biogenic methods are mainly limited to using microorganisms and plant extracts for synthesizing nanoparticles. However, there have been few studies conducted on animal secretions, and even fewer on mollusk secretions. This work focuses on the synthesis and characterization of copper oxide nanoparticles using purified (after 24-hour dialysis and concentration with a 20 kDa membrane) garden snail mucus.

The article needs to increase the evidence base in order to be published in the journal "Pharmaceutics".                                   I have highlighted some comments and suggestions below that might help the authors:

  1. How quantitative is the reduction of the copper salt used (CuSO4), how was this controlled?

Answer:

The nanoparticle formation experiment was conducted by gradually adding 00 ml of 0.1 M CuSO4 to 100 ml of a snail mucus fraction with MW>20 kDa under continuous stirring. Best results are obtained after the 3-rd day. Thanks for the suggestion to control the change of Cu at different stages of NP formation.

Research continues by following the formation of CuNPs in the presence of ascorbic acid and changing other conditions. We will also include control of reducing the amount of copper salt (CuSO4) used.

  1. Are thereany chemical changes in the concentrated mucus fraction during nanoparticle synthesis? Why is the reduction of Cu ions occurring: what components of the mucus are providing this? What is the nature of the interaction between mucus components and Cu nanoparticles?

Answer:

During the synthesis of NPs, there is a change in the concentration of some proteins that are involved in their formation, as shown in Fig. 10, a comparative analysis of the liquid chromatograms of the eluted mucus before and after preparation of NPs and of the electrophoresis in Fig. 10.

The reduction of copper ions is related to their participation in the formation of NPs, and the unreacted ones are removed by washing the NPs. Based on the information about change  proteins concentration in the mixture, provided by ImageQuantTM TL analysis two proteins with MWs of 30.691 kDa and 26.549 kDa were identified, which were proposed to be involved in the synthesis of CuONPs-Muc. Based on published data on proteins contained in C. aspersum mucus, we can hypothesize that proteins with MW 26549 kDa and 30591 kDa most likely exhibit antioxidant activity and belong to the superoxide dismutase, glutathione peroxidase (GPx) and glutathione transferases (GST ).

Protein-nanoparticle interactions are very well described by Drozd et al., 2022.  During the formation of NPs, mainly two types of interactions are observed: between Cu ions and the proteins, as well as between the outer surfaces of the proteins themselves.

The following text has been added to the article line 478: " The formation of NPs is mainly the result of two types of interactions: between Cu ions and the side groups of proteins, as well as between the outer surfaces of the mucus proteins themselves. The formation of NPs is based on intermolecular forces such as van der Waals, electrostatic, covalent, hydrogen bonds, π-π arrangement [Drozd et al., 2022].

“Proteins are made up of amino acid residues that contain basic functional groups; amino-, hydroxyl-, carboxyl-group, indole ring, imidazole side chain, sulfhydryl and the interactions can be of different types (van der Waals, electrostatic, covalent, hydrogen bonds, π-π stacking). Covalent bonds of Cu(I) with methionine (Met), tyrosine (Tyr) and histidine (His) and electrostatic cation-π interactions with tryptophan (Trp) have been established in proteins. Mucus contains proteins that have different side chains and their interactions with Cu ions are covalent, most likely with both proteins.”

Drozd, M.; Duszczyk, A.; Ivanova, P.; Pietrzak, M. Interactions of proteins with metal-based nanoparticles from a point of view of analytical chemistry - Challenges and opportunities. Advances in Colloid and Interface Science 2022, 304, 102656

  1. Considering that the size of nanoparticles is essential for antimicrobial activity, it must be determined accurately. You can clearly determine the dimensions using TEM. From Fig. 5 obtained using SEM, it is not possible to accurately determine the sizes of individual particles. The indicated sizes of 70-200 nm most likely correspond not to copper oxide nanoparticles, but to a complex of nanoparticles with components of the mucus used for the synthesis.

Answer:

We thank the Reviewer for this valuable remark. The Reviewer is right, that TEM is the most “powerful” method for observation the shape and morphology of almost any kind of nanoparticles. And especially for the shape and morphology of individual nanoparticles. Our aim is to get the overall (general/average) evaluation of the size of the nanoparticles, thus we use SEM in order to accumulate large number of data. Although the nanoparticles are partially hindered, we used a large amount of SEM images (which allows us to extract clearly visible individual nanoparticles, not aggregates) in order to evaluate the size distribution. Let us point out, that the indicated size of the nanoparticles is between 125-150 nm, as it is seen from the highest histogram peak, with almost symmetrical decline in larger and lower sizes, while the values of 70 and 200 nm are far beyond the average size and could be considered as non-representative.     

  1. In the caption to Fig. 5 A): CuONPs-SS, what does SS mean? Fig. 5 B): The sizes of copper nanoparticles (CuNPs) rather than copper oxide (CuONPs) are analyzed.

Answer

A correction was made Fig. 5 A) and Fig. 5 B): with CuNPs-Muc indicated

  1. To avoid confusion, you need to bring the names of the resulting compounds into uniformity. It is more correct to call copper oxide nanoparticles with mucus components CuONPs-Muc.

Answer:

We made the corrections in the manuscript to copper oxide nanoparticles with mucus components CuONPs-Muc

  1. The conclusion that exposed indole tryptophan nuclei from the surface of proteins penetrated into the resulting CuO nanoparticles requires clarification. (167 stock)

Answer:

It is known that amino acids such as proline, tryptophan, or tyrosine are often used as ligands in nanoparticle preparation processes [46]. Due to their ability to interact with hydrophobic regions in protein secondary structures, they disrupt the natural tendency of selected proteins to form oligomers by stacking β-sheet structures [47]. Aggregation and fibrillation of proteins controlled by nanomaterials can contribute to a change in the behavior of proteins, which is expressed in a change in fluorescence intensity or a shift of the fluorescence emission from some NPs as a result of interaction with proteins [48]. The observed change in the fluorescence analysis of the mucus fraction with MW >20 kDa before and after the formation of CuONPs-Muc is due to the formation of a protein corona, which leads to a change in the conformation and environment of the tryptophan residues in the proteins.

Tryptophan residues located on the surface of proteins in the mucus fraction with MW>20 kDa are responsible for the fluorescence emission at 351 nm (λex=295 nm). After the formation of biogenic nanoparticles, they find themselves in a more hydrophobic environment, buried in the protein corona of the formed CuONPs-Muc. As a result, a blue shift of the fluorescence emission at 341 nm (λex=295 nm) and a significant decrease in its intensity was observed. The presented fluorescence study shows the reorganization of secondary structure and spatial conformation (or three-dimensional conformation) of proteins in the mucus fraction with MW>20 kDa, supporting the formation of CuONPs-Muc, as reducing and stabilizing agents.

The fluorescence emission observed after excitation at 280 nm shows a predominance of tryptophan and tyrosine residues in mucus proteins with MW>20 kDa before and after the formation of the biogenic nanoparticles - CuONPs-Muc. In this case, the same trend as for excitation at 295 nm was observed, but the detected blue shift of the fluorescence emission maximum was greater from 351 nm to 316 nm.

Additional explanation is included on line 447-473.

Wang, S.; Zheng, J.; Ma, L.; Petersen, R.B.; Xu, L.; Huang, K. Inhibiting protein aggregation with nanomaterials: the underlying mechanisms and impact factors. Biochim Biophys Acta-Gen Subj 2022, 1866(2), 130061. https://doi.org/10.1016/j. bbagen.2021.130061

John, R.; Mathew, J.; Mathew, A.; Aravindakumar, C.T.; Aravind, U.K. Probing the Role of Cu(II) Ions on Protein Aggregation Using Two Model Proteins. ACS Omega. 2021, 6(51), 35559-35571. doi: 10.1021/acsomega.1c05119.

  1. Requires more detailed explanation - “more favorable intramolecular interactions in the protein” (line 277).

Answer

The information presented in Figure 9 and Table 1, regarding the interactions between the protein and CuONPs-Muc, as well as its thermal stability, from TG-DSC analyses, shows a significant difference in the behavior of the two samples. An explanation for the faster decomposition of NPs compared to pure mucus may be due to the adsorbed low molecular weight compounds from the mucus that form CuONPs.

In addition, the larger enthalpy values (ΔH) determined from the peak area of the DSC curves for the pure mucus sample (ΔH= -1216J/g) reflect stronger intramolecular interactions in the proteins.

This difference can be explained by the proteins in the mucus having different molecular weights. Protein size mainly affects the conformational change, folding and unfolding of proteins, and also the surface coverage of NPs. Proteins can be divided into three subcategories: small "hard" proteins with a rigid structure, large "soft" proteins with a flexible structure, and medium proteins. Small proteins do not undergo conformational changes upon contact with the substrate surface. Whereas, larger proteins undergo greater conformational changes caused by their contact with the NP surface [Andrade et al.,1992]. Therefore, the lower thermodynamic affinity of "stiff" proteins than that of larger proteins can be explained by the weaker conformational changes of the structure, reflected in a smaller number of binding points per protein molecule [Dee et al., 2002].

The processes of degradation of protein molecules, which are endothermic, in the composite are associated with the removal of low molecular weight compounds from the surface of the copper-containing particles (van der Waals forces). This is the reason for the less pronounced endothermic peak and the more pronounced mass loss (Mloss=69.44%) of the composite sample up to 130°C. After the separation of the low-molecular compounds, complete destruction of the material occurs.

The output mucus contains proteins of low and high molecular weight. In high-molecular proteins, the interactions between the side chains in the interior of the molecule are more pronounced, which is related to the higher conformational stability compared to low-molecular proteins. Therefore, the larger enthalpy values (ΔH) determined from the peak area of the DCS curves for the pure mucus sample (ΔH= -1216J/g) reflect stronger intramolecular interactions in the protein (Table 3).

Andrade, J.D.; Hlady, V.; Wei, A.P. Adsorption of complex proteins at interfaces. Pure Appl Chem 1992, 64, 1777–81.

Dee, K.C.; Puleo, D.A.; Bizios, R. Chapter 3 - Protein-surface interactions. An Introduction to Tissue-Biomaterial Interactions. John Wiley & Sons, Inc. 2002, p. 37–52.

  1. The first mention of the latin name of microorganism in the text should becomplete, not abbreviated.

Answer:

We fully agree with your suggestion and have made the necessary corrections in the manuscript.

  1. In the text on line 361, it is stated that vancomycin (V) and erythromycin (E) are most effective against gram-positive bacteria. However, Fig. 14A shows that cephalexin (CN) has greater antibiotic activity against three out of four strains, whereas erythromycin is only effective against one out of four.

Answer:

Thank you for your remark. However, Vancomycin and Erythromycin are more effective against Gram + bacteria (B. subtilis, B. spizizenzii, S. aureus and L. innocula), while against Gram – bacteria (E. coli, S. thyphimurium, S. eneritidis and S. maltophyla) more effective is the Vancomycin. The confusion arises from a technical mistake – the designations in figure A and B are reversed.

  1. The abstract, keywords, and conclusions of the article discuss fungal strains Candida albicans ATCC 10231 and Saccharomyces cerevisiae NBIMCC 584, but the text of the paper does not contain information on experiments or their results with these organisms.

Answer:

Candida albicans and Saccharomyces cerevisiae names have been omitted from the abstract and key words

Reviewer 2 Report

Comments and Suggestions for Authors

The authors developed a new complex of CuO nanoparticles from mucus of garden snail Cornu aspersum and evaluate its antimicrobial activity. Besides, many analytical techniques were applied for characterization for the novel complex, such as: UV-Vis, high performance liquid chromatography (HPLC), 1D-polyacrylamide gel electrophoresis (1D-PAGE), Fourier transform infrared spectroscopy (FTIR), scanning electron microscopy (SEM), Raman spectroscopy imaging,  and thermogravimetric analyzes (TG-DSC). However, many comments should be addressed and carefully revised.

1-     In the abstract, write full name for UV-Vis as all others methods

2-     Write full names for all abbreviations when first mentioning e.g.

A.     Write full name for BCA in line 110. Also for MALDI-MS, line 123

B.     In line 174, write full name for EDS

C.     In line 281, write full name for 1D-PAGE

D.     In line 318, write full name for TM TL

3-     The abstract should contain numerical values in the results subsection

4-     In the abstract, name of strains of Gram-positive (Gram+) and Gram-negative (Gram–) bacteria are required.

5-     In key words, please remove ATCC 3310231 for better homogenization

Candida albicans ATCC 3310231 and Saccharomyces cerevisiae

6-     The following article should be cited as they discussed similar ideas but for silver nanoparticles

https://doi.org/10.1038%2Fs41598-021-92478-4

Terrestrial snail-mucus mediated green synthesis of silver nanoparticles and in vitro investigations on their antimicrobial and anticancer activities

7-     The following review article should be cited too

https://doi.org/10.1002/mbo3.1263

Microbial diversity of garden snail mucus

8-     For line 47, add one or two review articles

Castillo-Henríquez, L., Alfaro-Aguilar, K., Ugalde-Álvarez, J., Vega-Fernández, L., Montes de Oca-Vásquez, G. and Vega-Baudrit, J.R., 2020. Green synthesis of gold and silver nanoparticles from plant extracts and their possible applications as antimicrobial agents in the agricultural area. Nanomaterials10(9), p.1763.

Rónavári, A., Igaz, N., Adamecz, D.I., Szerencsés, B., Molnar, C., Kónya, Z., Pfeiffer, I. and Kiricsi, M., 2021. Green silver and gold nanoparticles: Biological synthesis approaches and potentials for biomedical applications. Molecules26(4), p.844.

9-     In the introduction, the information about toxicity of  in vivo intake of garden snail mucus should be illustrated as the authors recommend its efficiency in the form of the new Nano complex as antimicrobial agent 

10-  Add reference for line 100, using a patented technology

11-  In line 109, state why Bradford method is more sensitive than Lowry and BCA ?

12-  In line 135 and 136 add reference for the synthesis method

13-  In figure 3, the authors should explain why the absorbance was decreased after 3 days ? is the synthesized CuONPs was corrupted ?

14-  In figure 10 B, which HPLC chromatogram is for mucus with MW > 20 kDa and which is for the supernatant after centrifugation of CuONPs, separated on a C18 column (8.00 x 250 mm)  ? please indicate that in the figure

15-  Colured figure 12 and 13 will be superior

16-  In figure 13 and 14, Vancomycin (V) should be written as Vancomycin (VA) as in the figure . also in line 362

17-  Lines 422- 424, needs appropriate reference

18-  Short paragraph about efficiency of the new synthesized CuONPs  against gram –ve bacteria should be stated.

19-  In line 410, 5 days or 3 days is optimal for the complex formation ?

20-  Line 686, needs appropriate reference for CLSI M39 Guidelines

21-  Future plans should be illustrated more

Best wishes

Author Response

We would like to thank the reviewers and the Editor for the time and effort taken to improve our manuscript “Development of CuO nanoparticles from mucus of garden snail Cornu aspersum as new antimicrobial agents” Pavlina Dolashka, Karina Marinova, Petar Petrov, Ventsislava Petrova, Bogdan Ranguelov, Stela Atanasova-Vladimirova, Dimitar Kaynarov, Ivanka Stoycheva, Emiliya Pisareva, Anna Tomova, Angelina Kosateva, Lyudmila Velkova, and Alexander Dolashki.

We have revised the manuscript according to all comments. We appreciate the opinion of the reviewers and agree with their comments. The authors have carefully considered the comments and tried our best to address every one of them. We hope the manuscript, after careful revisions, meets your high standards. The revised parts are marked in bold in the revised manuscript.

The authors developed a new complex of CuO nanoparticles from mucus of garden snail Cornu aspersum and evaluate its antimicrobial activity. Besides, many analytical techniques were applied for characterization for the novel complex, such as: UV-Vis, high performance liquid chromatography (HPLC), 1D-polyacrylamide gel electrophoresis (1D-PAGE), Fourier transform infrared spectroscopy (FTIR), scanning electron microscopy (SEM), Raman spectroscopy imaging, and thermogravimetric analyzes (TG-DSC). However, many comments should be addressed and carefully revised.

  1. In the abstract, write full name for UV-Vis as all others methods

Answer:

We have corrected the abstract and added the full name for UV-Vis as all other methods: “Various methods and techniques were applied, such as: ultraviolet–visible spectroscopy (UV-Vis), high performance liquid chromatography (HPLC), one dimensional polyacrylamide gel electrophoresis (1D-PAGE), Fourier transform infrared spectroscopy (FTIR), scanning electron microscopy combined with energy dispersive spectroscopy  (SEM/EDS), Raman spectroscopy and imaging, thermogravimetric analyzes (TG-DSC), etc.”

  1. Write full names for all abbreviations when first mentioning e.g.

Answer: Thank you for the recommendation. This has already been done.

  1. Write full name for BCA in line 110. Also for MALDI-MS, line 123

Answer: We have added the full name for BCA – “bicinchoninic acid (BCA) protein assay” and for MALDI-MS – “Matrix-assisted laser desorption/ionization time-of-flight mass spectrometry (MALDI-Tof-MS)”

  1. In line 174, write full name for EDS.

Answer: We have made the necessary correction as follows (in line.174.):

“2.2.3. Characterization of the obtained CuONPs-Muc by scanning electron microscopy combined with energy dispersive spectroscopy (SEM/EDS)”

  1. In line 281, write full name for 1D-PAGE

Answer: We have added full name for 1D-PAGE one dimensional -polyacrylamide gel electrophoresis

  1. In line 318, write full name for TM TL

Answer:

We corrected in line 281, 2.4.2. ImageQuant TM TL as follows:

“2.4.2. Image Analysis of 10% SDS-PAGE with ImageQuant™ TL v8.2.0 Software.”

  1. The abstract should contain numerical values in the results subsection.

Answer: We have corrected the Abstract as follows:

Abstract: Several biologically active compounds involved in the green synthesis of silver and gold nanoparticles have been isolated and characterized from snail mucus. This paper presents a successful method for the application of snail mucus from Cornu aspersum as a bioreducing agent of copper sulfate and as a biostabilizer of the obtained copper oxide nanoparticles (CuONPs-Muc). The synthesis at room temperature and neutral pH formed nanoparticles with a spherical shape and an average diameter of 150 nm. The structure and properties of CuONPs-Muc have been characterized by various methods and techniques, such as: ultraviolet-visible spectroscopy (UV-Vis), high-performance liquid chromatography (HPLC), one-dimensional polyacrylamide gel electrophoresis (1D-PAGE), up-conversion infrared spectroscopy Fourier transform (FTIR), scanning electron microscopy combined with energy dispersive spectroscopy (SEM/EDS), Raman spectroscopy and imaging, thermogravimetric analyzes (TG-DSC), etc. Mucus proteins with molecular weight of 30.691 kDa and 26.549 kDa were identified, which are involved in the biogenic production of CuONPs-Muc. The formed macromolecular shell of proteins around the copper ions contributes to a higher efficiency of the synthesized CuONPs-Muc in inhibiting the bacterial growth of several Gram-positive (B. subtilis NBIMCC2353, B. spizizenii ATCC 6633, S. aureus ATCC 6538, L. innocua NBIMCC8755) and Gram-negative (E. coli ATCC8739, S. ente-itidis NBIMCC8691, S. typhimurium ATCC 14028, S. maltophilia ATCC 17666) bacteria compared to baseline mucus. The presented bioorganic synthesis of snail mucus provides CuONPs-Muc with a highly pronounced antimicrobial effect. These results will expand the knowledge in the field of natural nanomaterials and their role in emerging dosage forms.

  1. 4. In the abstract, name of strains of Gram-positive (Gram+) and Gram-negative (Gram–) bacteria are required.

Answer:  The names of Gram + and Gram – strains have been added at the abstract.

  1. 5. In key words, please remove ATCC 3310231 for better homogenization Candida albicans ATCC 3310231 and Saccharomyces cerevisiae

Answer:   Candida albicans and Saccharomyces cerevisiae names have been omitted from the abstract and key words.

  1. 6. The following article should be cited as they discussed similar ideas but for silver nanoparticles https://doi.org/10.1038%2Fs41598-021-92478-4

Terrestrial snail-mucus mediated green synthesis of silver nanoparticles and in vitro investigations on their antimicrobial and anticancer activities

Answer:

The indicated article is cited both in 1. Introduction, line 72-74 (“Also, ecofriendly synthesized biogenic AgNPs-SM from the mucus of Achatina fulica snail by Mane et al., showed strong antibacterial, antifungal as well as anticancer activity against Hela cells [21].”) and in the 3. Discussion (line 589). In References it is under No:

  1. Mane, P.C.; Sayyed, S.A.R.; Kadam, D.D.; et al. Terrestrial snail-mucus mediated green synthesis of silver nanoparticles and in vitro investigations on their antimicrobial and anticancer activities. Sci Rep 11, 13068 (2021).

  1. 7. The following review article should be cited too https://doi.org/10.1002/mbo3.1263, Microbial diversity of garden snail mucus.

Answer:

The mentioned article was cited under number 25, and after the corrections, it is under No 27 in References:

  1. Belouhova, M.; Daskalova, E.; Yotinov, I.; Topalova, Y.; Velkova, L.; Dolashki, A.; Dolashka, P. Microbial diversity of garden snail mucus. Microbiologyopen 2020, 11(1), e1263.

  1. 8. For line 47, add one or two review articles:

Castillo-Henríquez, L., Alfaro-Aguilar, K., Ugalde-Álvarez, J., Vega-Fernández, L., Montes de Oca-Vásquez, G. and Vega-Baudrit, J.R., 2020. Green synthesis of gold and silver nanoparticles from plant extracts and their possible applications as antimicrobial agents in the agricultural area. Nanomaterials10(9), p.1763.

Rónavári A, Igaz, N., Adamecz, D.I., Szerencsés, B., Molnar, C., Kónya, Z., Pfeiffer, I. and Kiricsi, M., 2021. Green silver and gold nanoparticles: Biological synthesis approaches and potentials for biomedical applications. Molecules26(4), p.844.

Answer:

Thank you for the suggestion. The mentioned two articles are cited in 1. Introduction (in line 49, line 49-54). They are under Nos 7 and 9 in References:

 Castillo-Henríquez, L.; Alfaro-Aguilar, K.; Ugalde-Álvarez, J.; Vega-Fernández, L.; Montes de Oca-Vásquez, G.; Vega-Baudrit, J.R. Green Synthesis of Gold and Silver Nanoparticles from Plant Extracts and Their Possible Applications as Antimicrobial Agents in the Agricultural Area. Nanomaterials 202010, 1763.

Rónavári, A.; Igaz, N.; Adamecz, D.I.; Szerencsés, B.; Molnar, C.; Kónya, Z.; Pfeiffer, I.; Kiricsi, M. Green Silver and Gold Nanoparticles: Biological Synthesis Approaches and Potentials for Biomedical Applications. Molecules 202126, 844. 

In 1. Introduction: “Silver (AgNPs) and gold (AuNPs) nanoparticles synthesized using plant extracts are of great interest among researchers [8,9]. Recently, Castillo-Henríquez et al., 2020 reported the green synthesis of gold and silver nanoparticles from plant extracts and their capacity as antimicrobial agents in the field of agriculture to combat bacterial and fungal pathogens. Moreover, this work makes a brief review of nanoparticles’ contribution to water treatment and the development of “environmentally-friendly” nanofertilizers, nanopesticides, and nanoherbicides [8].”

  1. 9. In the introduction, the information about toxicity of in vivo intake of garden snail mucus should be illustrated as the authors recommend its efficiency in the form of the new Nano complex as antimicrobial agent

Answer:  Thank you for the suggestion. We have added the information about the toxicity of garden snail mucus (line 92-104)

“Several results have been published on the toxicity of H. aspersa mucus extracts against eukaryotic cells. Trapella et al., reported a lack of toxicity and a proven cytostatic effect at all concentrations tested in vitro of mucus extracts (HelixComplex) on human dermal fibroblasts (MRC-5)[28]. This is also confirmed by a study by Gentili et al., 2020 on the non-toxic effect of HelixComplex on human keratinocytes [29].

Mucus has been found to protect cells from apoptosis and significantly induce cell proliferation and migration through direct and indirect mechanisms. Research by Deng et al. showed strong adhesion to wet tissue of mucus extracts from two land snails, hemostatic effect, good biocompatibility, and hemocompatibility, and in vivo pro-healing activity for skin wounds [30]. The mucus of the land snail H. aspersa was found to play a crucial role as a stabilizing agent in the synthesis of bioactive AgNPs [31]”

  1. 10. Add reference for line 100, using a patented technology.

Answer: We have added the necessary reference.

“This study analyzed an extract of mucus collected from snails C. aspersum, grown in Bulgarian eco-farms, using a patented technology [33].”

  1. Dolashka et al., BG Useful model 2097, 2015.

  1. 11. In line 109, state why Bradford method is more sensitive than Lowry and BCA?

Answer:

Bradford method for the quantification of protein concentration, is more sensitive (1-20 µg/mL) than BCA (25-2000 µg/mL). The key difference between Bradford and Lowry Protein Assay lies on the colorimetric technique they use. Bradford protein assay uses the Coomassie brilliant blue G-250 while Lowry protein assay uses copper ions (Cu+) ions and Folin–Ciocalteu reagent. Furthermore, the Bradford method gives quick results than the Lowry protein assay. However, both methods are highly sensitive methods and are subject to interference from various substances.

We have corrected the sentence as follows:

"After applied of the more sensitive Bradford method, than bicinchoninic acid (BCA) protein assay, a total protein concentration of 1.56 mg/mL in the obtained snail mucus was determined."

  1. 12. In line 135 and 136 add reference for the synthesis method.

Answer: We have not added a reference to the synthesis method since it was developed by us.

  1. 13. In figure 3, the authors should explain why the absorbance was decreased after 3 days? is the synthesized CuONPs was corrupted?

Answer:  

The synthesis of CuONPs-Muc was followed for 5 days. The visible color change of the CuSO4 solution from pale blue to dark green indicated that the formation of CuONPs-Muc started 2 days after mixing, becoming more intense with time until the 3rd day.

Figure 3 shows the UV-Vis spectra of the obtained synthesized CuONPs-Muc at different reaction times: after 24 hours (in green), after 48 hours (in light blue) and after 72 hours (in red). The curves in black and dark blue of Figure 3 shows the absorption spectra after centrifugation at 4000 rpm and 12000 rpm for 10 minutes, after 1 hour of synthesis.

  1. 14. In figure 10 B, which HPLC chromatogram is for mucus with MW > 20 kDa and which is for the supernatant after centrifugation of CuONPs, separated on a C18 column (8.00 x 250 mm)? Please indicate that in the figure.

Answer:

Chromatograms of mucus with MW > 20 kDa (black line) and the supernatant after centrifugation of CuONPs-Muc (green line), separated on a C18 column are presented in different colors, respectively.

  1. 15. Colored figure 12 and 13 will be superior

Answer: Figures 12 and 13 are provided in color.

  1. 16. In figure 13 and 14, Vancomycin (V) should be written as Vancomycin (VA) as in the figure, also in line 362

Answer: The abbreviation of Vancomycin (VA) has been corrected.

  1. 17. Lines 422- 424, needs appropriate reference.

Answer: We have added the necessary reference. It is under No 49 in References:

  1. Royer C.A. Probing protein folding and conformational transitions with fluorescence. Chem Rev. 2006, 106(5), 1769-84. doi: 10.1021/cr0404390.

  1. 18. Short paragraph about efficiency of the new synthesized CuONPs against gram –ve bacteria should be stated.

Answer: A short paragraph about efficiency of the new synthesized CuONPs-Muc against Gram – bacteria have been added.

  1. 19. In line 410, 5 days or 3 days is optimal for the complex formation?

Answer:  The optimal time for the complex formation of CuONPs-Muc was 3 days.

  1. 20. Line 686, needs appropriate reference for CLSI M39 Guidelines

Answer:

We have added the necessary reference. It is under No 83 in References:

  1. 83. Clinical and Laboratory Standards Institute. Analysis and presentation of cumulative antimicrobial susceptibility test data, 5th ed. Approved guideline. Clinical and Laboratory Standards Institute, Wayne, PA 2022.

  1. Future plans should be illustrated more

Answer:  The conclusion was rewritten :

The present research presents for the first time an environmentally friendly method without toxic waste for synthesizing biogenic CuONPs-Muc from the mucus fraction with MW>20 kDa of garden snail C. aspersum. The biosynthesized CuONPs-Muc were proven and characterized by applying various modern methods (UV-Vis and fluorescence spectroscopy, RP-HPLC analyses, electrophoretic analyses, mass spectrometry, SEM and EDS, FTIR and TG-DSC analyses). The CuОNP-Muc showed higher inhibition of the bacterial growth of various Gram+ (B. subtilis NBIMCC2353, B. spizizenii ATCC 6633, S. aureus ATCC 6538, L. innocua NBIMCC8755) and Gram– (E coli ATCC8739, S. enteritidis NBIMCC8691, S. typhimurium ATCC 14028, S. maltophilia ATCC 17666) bacteria compared to the initial mucus fraction with MW >20 kDa.

The results of the in vitro studies show that a combination of new biomolecules with antimicrobial properties from the mucus of the snail C. aspersum and the ecological approach – biological synthesis of metal nanoparticles can lead to new effective antimicrobial therapeutics against high-risk pathogens.

Research in this area will continue to obtain and characterize biogenic metal nanoparticles synthesized from the mucus of the snail C. aspersum under different conditions, including in the presence of ascorbic acid, and their biomedical application. To elucidate the mechanism of the antimicrobial activity of the obtained new biogenic nanoparticles, the cellular redox status, the level of transcription and the levels of apoptotic cell markers in the tested pathogenic microorganisms will be followed before and after inoculation with new NPs-Muc.

Comments and Suggestions for Authors and our answers

Reviewer 3 Report

Comments and Suggestions for Authors

The above manuscript approaches an interesting subject, a green procedure for the synthesis of CuONPs using snail slime. Some improvements should be undertaken:

-line 188: Fig 6. EDS spectra of Cu/Cu2O. It was not mentioned that maybe besides CuONPs, copper in other valence states could be formed. So an XRD analysis would solve this problem

-line 227-229: 'Cu(I)-O particles with the vibrational band of Cu2O... confirming the presence of calcined Cu nanoparticles '. More details about the resulted copper species. How were the nanoparticles clacined ?

-line 240-241: the processes occurring over time traced by DTG or DSC curves ?

-line 242: 'TG and DSC results presented in fig. 9 and table 2'; table 3 is missing

-the graphics in fig. 9 are not very clearly represented

-In Fig 14 B and A,  vancomycin is abbreviated VA instead of V (line 362 in the text)

-line 385-386: 'the purity of the sample from impurities'. To what impurities eliminated by dialysis does the authors refer to and what does the purity of the sample means ? 

-line 401: 'Another published information related to the advantage of using the whole plant extract'....What is the reference and the mentioned plant extract and what is its relevance to the subject ?

Author Response

We would like to thank the reviewers and the Editor for the time and effort taken to improve our manuscript “Development of CuO nanoparticles from mucus of garden snail Cornu aspersum as new antimicrobial agents” Pavlina Dolashka, Karina Marinova, Petar Petrov, Ventsislava Petrova, Bogdan Ranguelov, Stela Atanasova-Vladimirova, Dimitar Kaynarov, Ivanka Stoycheva, Emiliya Pisareva, Anna Tomova, Angelina Kosateva, Lyudmila Velkova, and Alexander Dolashki.

We have revised the manuscript according to all comments. We appreciate the opinion of the reviewers and agree with their comments. The authors have carefully considered the comments and tried our best to address every one of them. We hope the manuscript, after careful revisions, meets your high standards. The revised parts are marked in bold in the revised manuscript.

Reviewer 3

The above manuscript approaches an interesting subject, a green procedure for the synthesis of CuONPs using snail slime. Some improvements should be undertaken:

  1. Line 188: Fig 6. EDS spectra of Cu/Cu2O. It was not mentioned that maybe besides CuONPs, copper in other valence states could be formed. So an XRD analysis would solve this problem.

Answer:

We thank the Reviewer for this valuable comment. Indeed, besides CuONPs-Muc, copper in other valence states could be formed, we agree. XRD analysis will be included in all further investigations.

  1. Line 227-229: 'Cu(I)-O particles with the vibrational band of Cu2O... confirming the presence of calcined Cu nanoparticles '. More details about the resulted copper species. How were the nanoparticles clacined?

Answer:  

The text was corrected  ”Other important bands are observed at 661 and 616 cm-1 due to the stretching of Cu(I)–O particles, with the vibrational band of CuO at 616 cm-1 confirming the presence of Cu nanoparticles”.  

  1. 3. Line 240-241: the processes occurring over time traced by DTG or DSC curves?

Answer: The processes occurring over time traced by DTG and DSC are presented after Table 3

  1. Line 242: 'TG and DSC results presented in fig. 9 and table 2'; table 3 is missing.

Answer: The text is corrected.

  1. The graphics in fig. 9 are not very clearly represented

Answer:  

Figure 9. TG-DSC analysis of pure mucus sample and CuONPs-Muc The graphics in Fig. 9 is corrected.

  1. In Fig 14 B and A, vancomycin is abbreviated VA instead of V (line 362 in the text)

Answer: The abbreviation of Vancomycin (VA) has been corrected.

  1. Line 385-386: 'the purity of the sample from impurities'. To what impurities eliminated by dialysis does the authors refer to and what does the purity of the sample means?

  The UV-Vis spectra presented in figure 1 of snail mucus before and after dialysis confirm the purity of the sample from impurities “

Answer:

Before obtaining the mucus sample with MT>20 kDa, the starting mucus is dialyzed because it contains various low-molecular-weight impurities stabilizing reagents. The UV-Vis spectrum presented in Figure 1 of snail mucus shows high absorption due to these impurities and the proteins in the mucus. In order to synthesize NPs, the sample is purified by dialysis from these impurities so that only the proteins in the mucus with MT>20 kDa remain. The UV-Vis spectrum (Figure 1) after dialysis differs from the spectrum of the starting fraction, showing a typical protein UV-Vis spectrum with a maximum at 280 nm, thus confirming the purity of the sample from impurities.

  1. Lne 401: 'Another published information related to the advantage of using the whole plant extract'.... What is the reference and the mentioned plant extract and what is its relevance to the subject?

 „Another published information related to the advantage of using the whole plant extract was used in CuONPs synthesis, where all compounds in the extract participate in the process as reducing agents but also as capping agents in complex systems.“

Answer:

It has been found that the use of the whole plant extract has a number of advantages in the formation of NPs compared to the isolation and use of individual components of the extract. Based on this information, in our studies we have used a fraction that contains a wide range of proteins rather than individual proteins.

Reviewer 4 Report

Comments and Suggestions for Authors

Here, the authors synthesized CuO nanoparticles from the mucus of garden snail and evaluated its antimicrobial activity. The manuscript is decently written. Please address the following concerns before publication of this manuscript.

1.            Please correct the UV-Vis spectrum. It should be Absorbance vs. Wavelength

2.            From the SEM image, the nanoparticles are agglomerated and are not spherical. The size determination seems misleading.

3.            Authors claim the formation of CuONPs and CuNPs in the conclusions section. Please correct it.

4.            UV-Vis spectra cannot be used to ascertain the purity of a sample as the authors claim.

5.            Line 90 “CuONPs are among the group of oligodynamic noble metals” There should be a citation for this statement.

6.            Authors are confused between CuNPs and CuONPs throughout the manuscript.

7.            UV-Vis, fluorescence spectroscopy, RP-HPLC, electrophoretic analyses, MALDI, mass spectrometry, SEM and EDS, FTIR, or TG-DSC analyses cannot conclusively confirm the formation of CuO.  XRD analysis must be performed to validate the authors' claim of CuONPs.

Comments on the Quality of English Language

Moderate editing required

Author Response

We would like to thank the reviewers and the Editor for the time and effort taken to improve our manuscript “Development of CuO nanoparticles from mucus of garden snail Cornu aspersum as new antimicrobial agents” Pavlina Dolashka, Karina Marinova, Petar Petrov, Ventsislava Petrova, Bogdan Ranguelov, Stela Atanasova-Vladimirova, Dimitar Kaynarov, Ivanka Stoycheva, Emiliya Pisareva, Anna Tomova, Angelina Kosateva, Lyudmila Velkova, and Alexander Dolashki.

We have revised the manuscript according to all comments. We appreciate the opinion of the reviewers and agree with their comments. The authors have carefully considered the comments and tried our best to address every one of them. We hope the manuscript, after careful revisions, meets your high standards. The revised parts are marked in bold in the revised manuscript.

Reviewer 4

Comments and Suggestions for Authors

Here, the authors synthesized CuO nanoparticles from the mucus of garden snail and evaluated its antimicrobial activity. The manuscript is decently written. Please address the following concerns before publication of this manuscript.

  1. Please correct the UV-Vis spectrum. It should be Absorbance vs. Wavelength.

Answer:

Thank you for the remark. We have made a correction.

  1. From the SEM image, the nanoparticles are agglomerated and are not spherical. The size determination seems misleading.

Answer:

We thank the Reviewer for this valuable remark. In order to get the overall (general/average) evaluation of the size of the nanoparticles we use SEM and we accumulate large number of data for post processing. Although the nanoparticles are partially hindered, we used a large amount of SEM images (which allows us to extract clearly visible individual nanoparticles, not aggregates) in order to evaluate the size distribution. It is clear that the shape of the nanoparticles is not spherical and in order to get a not misleading size distribution we have measured the dimensions in two perpendicular directions and take the average, thus we are sure to approach the right size(s) of the nanoparticles. Let us point out, that the indicated size of the nanoparticles is between 125-150 nm, as it is seen from the highest histogram peak.

  1. Authors claim the formation of CuONPs and CuNPs in the conclusions section. Please correct it.

Answer:

To avoid confusion we named copper oxide nanoparticles with mucus components CuONPs-Muc. We synthesized biogenic nanoparticles from the mucus fraction with MW>20 kDa of garden snail C. aspersum - CuONPs-Muc.

  1. UV-Vis spectra cannot be used to ascertain the purity of a sample as the authors claim.

UV-Vis спектрите не могат да се използват за установяване на чистотата на пробата, както твърдят авторите.

Answer:

The UV-Vis spectra presented in Figure 1 show the differences between a sample of crude slime extract with MW>20 kDa and a sample containing slime extract with MW>20 kDa after removal of unwanted low molecular weight impurities by dialysis followed by concentration on a 20 kDa membrane. As shown in Figure 1, a difference was observed between the crude mucus extract and the protein fraction. The high absorption intensity of the crude clay extract is due to the presence of unwanted impurities of low molecular weight. The typical protein UV-Vis spectrum with a maximum at 280 nm was observed for protein fraction with MW>20 kDa from C. aspersum mucus.

  1. Line 90 “CuONPs are among the group of oligodynamic noble metals” There should be a citation for this statement.

 Answer: We have added the necessary reference.

  1. Woźniak-Budych, M.J.; Staszak, K.; Staszak, M. Copper and Copper-Based Nanoparticles in Medicine-Perspectives and Challenges. Molecules 2023, 28(18), 6687. doi: 10.3390/molecules28186687. PMID: 37764463; PMCID: PMC10536384.
  2. Authors are confused between CuNPs and CuONPs throughout the manuscript.

Авторите са объркани между CuNP и CuONP в целия ръкопис.

Answer: To avoid confusion we named copper oxide nanoparticles with mucus components CuONPs-Muc.

  1. UV-Vis, fluorescence spectroscopy, RP-HPLC, electrophoretic analyses, MALDI, mass spectrometry, SEM and EDS, FTIR, or TG-DSC analyses cannot conclusively confirm the formation of CuO.  XRD analysis must be performed to validate the authors' claim of CuONPs.

Answer:

Individual consideration of the obtained results from UV-Vis, fluorescence spectroscopy, RP-HPLC, electrophoretic analyses, MALDI, mass spectrometry, SEM and EDS, FTIR and TG-DSC analyzes cannot really confirm the formation of CuO definitively, but the complex analysis of the results , obtained by these independent studies, confirm the preparation of CuONPs-Muc.

The combined methods are presented in a review of Drozd ет. Ал. 2022., which describes the use of these methods to prove the properties and structure of NPs.

Drozd, M.; Duszczyk, A.; Ivanova, P.; Pietrzak, M. Interactions of proteins with metal-based nanoparticles from a point of view of analytical chemistry - Challenges and opportunities. Advances in Colloid and Interface Science 2022, 304, 102656

We agree that XRD analysis plays an important role in the characterization of nanoparticles, providing information on the crystal structure and size of the nanoparticles. We plan to use it in future research, as we do not have such an apparatus at this stage.

Round 2

Reviewer 2 Report

Comments and Suggestions for Authors

the authors did a great job in the revision process. the paper could be published in the current form 

Greetings 

Author Response

Comments and Suggestions for Authors

the authors did a great job in the revision process. the paper could be published in the current form.  Greetings 

NSWER :
I would like to thank REVIWER again for the suggestions and corrections.

Reviewer 3 Report

Comments and Suggestions for Authors

 I accept the manuscript after the revision and I think it is all right to publish it.

Author Response

Comments and Suggestions for Authors

 I accept the manuscript after the revision and I think it is all right to publish it.

ANSWER :
I would like to thank REVIWER again for the suggestions and corrections.

Reviewer 4 Report

Comments and Suggestions for Authors

Authors performed most of the suggested modifications

Comments on the Quality of English Language

Minor editing required

Author Response

Comments and Suggestions for Authors

Authors performed most of the suggested modifications

Comments on the Quality of English Language - Minor editing required

ANSWER :  I would like to thank REVIWER again for the suggestions and corrections.  We have corrected again the manuscript.